# Kin17 promotes rDNA transcription, ribosomal biogenesis, and cortical lamination

Wenbo Li [ID] [1,2], Juan Zhang [ID] [1,3,4]✉ & Qiang Liu [ID] [1,2,3,4,5]✉

## Abstract

**During brain development, neural progenitor cells (NPCs) undergo rapid division, necessitating efficient ribosomal biogenesis for proliferation. Yet, the regulatory mechanisms remain largely elusive. Here, we report that the DNA binding protein Kin17 exhibits development-dependent expression and plays a vital role in embryonic development. Complete loss of Kin17 in mice leads to embryonic lethality, while Kin17 depletion specifically in NPCs allows embryonic survival but results in reduced brain size and cortical lamination defects. Our findings demonstrate that these cortical malformation stems from impaired NPC proliferation and differentiation. Mechanistically, we show that Kin17 binds to the promoter region of rDNA, sequentially recruiting NCL and Polr1a, thereby promoting rDNA transcription. Consequently, Kin17 facilitates ribosome biogenesis and protein translation in NPCs. This study underscores a critical role of Kin17 in promoting rDNA transcription and ribosomal biogenesis in NPCs during brain development, which is essential for proper cortical lamination.**

**Keywords** Kin17; Neural Progenitor Cells; Cortical Lamination; rDNA Transcription; Ribosome Biogenesis
**Subject Categories** Chromatin, Transcription & Genomics; Neuroscience; Translation & Protein Quality

## Introduction

Neural progenitor cells (NPCs) are the primary precursor cells in the developing brain, giving rise to both glial and neuronal lineages (Liu et al, 2023). In early developmental stages, NPCs undergo proliferative division to expand their population (Katada et al, 2021; Sultan et al, 2018). As development progresses, NPCs differentiate into radial glial cells (RGs) and intermediate progenitors (IPs), which ultimately give rise to neurons (Fang et al, 2013). These neurons migrate to the cortical plate (CP) region, where they contribute to the formation of the six-layered structure of the cerebral cortex (Arai and Taverna, 2017). Both the proliferation and differentiation of NPCs are pivotal in determining cortical size and thickness (Chenn and Walsh, 2002; Sun and Hevner, 2014).

Protein synthesis is crucial for normal brain development and function (Kraushar et al, 2021; Zahr et al, 2018). As such, the regulation of protein biosynthesis machinery has emerged as a key mechanism controlling cellular proliferation and differentiation (Kristensen et al, 2013; Xu et al, 2016a). In mammals, ribosome biogenesis begins with the transcription of ribosomal DNA (rDNA) by RNA polymerase I (Pol I), producing the 45S precursor rRNA (45S pre-rRNA) (Colis et al, 2014; Sun et al, 2023). This pre-rRNA is subsequently cleaved and processed into mature rRNAs, including 5.8S, 18S, and 28S rRNA, which assemble with ribosomal proteins (RPs) to form functional ribosomes (Jiao et al, 2023). Disruption of Pol I subunits, such as Polr1a, in NPCs reduces rRNA synthesis and leads to developmental brain abnormalities (Falcon et al, 2022). However, how rDNA transcription is regulated in NPCs and its impact on brain development remains largely unresolved. Kin17 is a DNA binding protein highly conserved from yeast to humans (le Maire et al, 2006). It has been implicated in promoting DNA replication in cancer cells and is essential for embryonic development in zebrafish and brain development in *Drosophila* (Connell et al, 2024; Liu et al, 2015; Zeng et al, 2011). However, its role in mammalian brain development remains unclear.

In this study, we investigated the role of Kin17 in mammalian cortical development. We observed that global depletion of Kin17 in mice leads to embryonic lethality, whereas conditional knockout of Kin17 specifically in NPCs (*Kin17* cKO) allows survival but results in severe cortical lamination defects. *Kin17* cKO mice display a significantly reduced cortex size and decreased cell numbers within each cortical layer, highlighting a critical role for Kin17 in proper cortical architecture and overall brain development. Our findings reveal that Kin17 deficiency impairs both proliferation and differentiation of NPCs, a phenotype also recapitulated in neurosphere cultures derived from *Kin17* cKO mice. Mechanistically, we show that Kin17 interacts with NCL, and this interaction is required to recruiting Polr1a to the rDNA promoter, thereby promoting rDNA transcription. In summary, our study identifies Kin17 as a key regulator of ribosomal biogenesis and NPC proliferation during brain development. By facilitating rDNA transcription through NCL and Polr1a

[1]Hefei National Laboratory for Physical Sciences at the Microscale, Division of Life Sciences and Medicine, University of Science and Technology of China, 230027 Hefei, China. [2]Department of Neurology, Institute on Aging and Brain Disorders, The First Affiliated Hospital of USTC, University of Science and Technology of China, 230027 Hefei, China. [3]Anhui Province Key Laboratory of Biomedical Aging Research, University of Science and Technology of China, 230027 Hefei, China. [4]CAS Key Laboratory of Brain Function and Disease, University of Science and Technology of China, 230026 Hefei, China. [5]Center for Advanced Interdisciplinary Science and Biomedicine of IHM, Division of Life Sciences and Medicine, University of Science and Technology of China, Hefei, China. ✉E-mail: zj2014@ustc.edu.cn; liuq2012@ustc.edu.cn

recruitment, Kin17 ensures the proper formation of cortical layers and overall brain structure.

## Results

### Depletion of Kin17 in neural progenitor cells leads to brain developmental atrophy

To explore the role of Kin17 in mammalian embryonic development, we employed CRISPR/Cas9 technology to delete exon 5 of the *Kin17* locus, generating *Kin17* knockout (KO) mice (Fig. EV1A). While heterozygous *Kin17* KO mice (*Kin17*$^{+/-}$) were viable and fertile, no homozygous KO (*Kin17*$^{-/-}$) offspring were recovered (Fig. EV1B). Genotyping embryos produced from *Kin17*$^{+/-}$ crosses at embryonic day (E) 7.5, 13.5, and postnatal day (P) 0.5 confirmed the absence of *Kin17*$^{-/-}$ embryos (Fig. EV1B), indicating early embryonic lethality upon genetic depletion of Kin17.

Analysis of the EMBL-EBI Expression Atlas revealed high Kin17 expression in the brain (Moreno et al, 2022), which declines during embryonic development and further decreases postnatally (Fig. EV1C). Immunoblotting of brain tissue from wild-type (WT) mice from E10.5 to P63 similarly showed a progressive decline in Kin17 expression (Fig. 1A), supporting the involvement of Kin17 in brain development. By conducting immunofluorescence staining for Kin17 alongside the neuronal marker NeuN or the astrocytic marker GFPA in adult brain tissue, we observed a complete colocalization of Kin17 with NeuN and no colocalization of Kin17 with GFAP (Figs. 1B and EV1D). This indicates that Kin17 is specifically localized to neurons in the mature brain. By conducting immunostaining for Kin17 and neural stem marker Nestin in the cortices of E14.5 mice, we demonstrated the localization of Kin17 in neural stem cells (Fig. 1C). These lines of evidence prompt us to investigate its potential role in neuronal development.

To investigate Kin17's role in neuronal development, we conditionally deleted Kin17 in NPCs. We generated *Kin17* floxed (*Kin17*$^{flox/flox}$) mice by flanking exon 5 with loxP sites and crossed these mice with *Nestin-Cre* mice, which express Cre recombinase driven by the Nestin promoter, resulting in the production of *Kin17*$^{flox/flox}$/*Nestin-Cre* mice (*Kin17* cKO mice) (Fig. 1D). Efficient Kin17 deletion in cKO brains was confirmed by immunoblotting and qPCR analyses (Figs. 1E and EV1E). While viable *Kin17* cKO embryos from crossing of *Kin17*$^{flox/flox}$ and *Kin17*$^{flox/+}$/*Nestin-Cre* mice were present at Mendelian ratios, no viable *Kin17* cKO mice were observed postnatally (Fig. EV1F), suggesting survival through embryogenesis but postnatal lethality.

Considering that Nestin expression begins around E10 (Lee et al, 2012; Oldrini et al, 2018), prompting us to examine Kin17 expression in the brain of Kin17 cKO embryos starting at E12.5. We observed a significant reduction in Kin17 expression in Kin17 cKO brains at E12.5 and a further reduction at E14.5, compared to their respective controls (Fig. 1F). The Kin17 expression in Kin17 cKO brains remained consistently lower than controls through E16.5 and E18.5 (Fig. 1F).

To assess the impact of Kin17 on mouse brain development, we measured brain weights at E12.5, E14.5, E16.5, and E18.5, and detected no significant differences between control and Kin17 cKO embryos at E12.5 or E14.5 (Fig. 1G). However, by E16.5 and E18.5,

Kin17 cKO embryos exhibited a marked reduction in brain weight compared to controls (Fig. 1G). These findings indicate that depletion of Kin17 in NPCs results in brain atrophy, which manifests as early as E16.5. Supporting this observation, cortical area measurements at E16.5 showed significantly smaller cortices in Kin17 cKO embryos compared to controls (Fig. 1H,I). Collectively, these results demonstrate that Kin17 is essential for normal brain development, with its depletion in NPCs leading to progressive brain atrophy.

### Depletion of Kin17 in NPCs leads to defective cortex lamination

To investigate how Kin17 loss contributes to brain atrophy, we performed Nissl staining on E16.5 brains. This analysis revealed a significant reduction in total neuronal numbers in the cortex of *Kin17* cKO mice compared to controls (Fig. 2A). Additionally, cortex thickness was significantly decreased in *Kin17* cKO mice at E16.5 compared to controls (Fig. 2B).

The cerebral cortex consists of six stacked layers (I–VI), arranged from the superficial to the deep region of the cortex (Molyneaux et al, 2007). Immunostaining indicated that Kin17 is ubiquitously expressed in all cortical layers of wild-type brains during embryo development (Fig. EV2A,B). To assess the impact of Kin17 on cortical organization, we conducted immunofluorescence staining using antibodies specific to layer marker proteins. We observed a notable reduction in the expression of Tbr1 (a specific marker for layer VI), Bcl11b (a specific marker for layer V), and Satb2 (a marker for layers II–IV) in Kin17 cKO cortices at E16.5, compared to controls (Fig. 2C–F). These decreases indicate thinning across multiple cortical layers. Interestingly, no difference in the expression of Tbr1 and Bcl11b was detected between control and *Kin17* cKO mice at E14.5 (Fig. EV2C–E), suggesting that lamination defects arise after this stage.

By assessing apoptosis using TUNEL staining, *Kin17* cKO mice showed a significantly higher number of TUNEL-positive cells in the cortical brain at E16.5 compared to control mice (Fig. 2G,H), indicating that defective neurons in the developing brain are prone to cell death. By conducting immunostaining in control and Kin17 cKO cortices, we found that NPM1 and NCL, both nucleolar marker proteins, remained localized within the nucleoli in Kin17 cKO cortices (Fig. 2I,J). Additionally, we examined p53 protein levels by conducting immunoblotting and detected no difference between control and Kin17 knockdown (KD) 293T cells. (Fig. EV2F). These findings indicate that no occurrence of nucleolar stress and the apoptosis observed in *Kin17* cKO mice proceeds independently of p53 signaling. Collectively, these findings demonstrate that Kin17 depletion disrupts normal cortical layer formation, contributing to cortical malformation and brain atrophy.

### Kin17 deficiency impairs NPC proliferation and differentiation

To investigate how Kin17 depletion in NPCs affects cortical lamination and brain development, we performed RNA sequencing (RNA-seq) on E16.5 cortices from both *Kin17* cKO and control mice. Our analysis revealed 945 differentially expressed genes (DEGs) (|Fold Change| ≥2 and FDR < 0.05), with 717 genes upregulated and 228

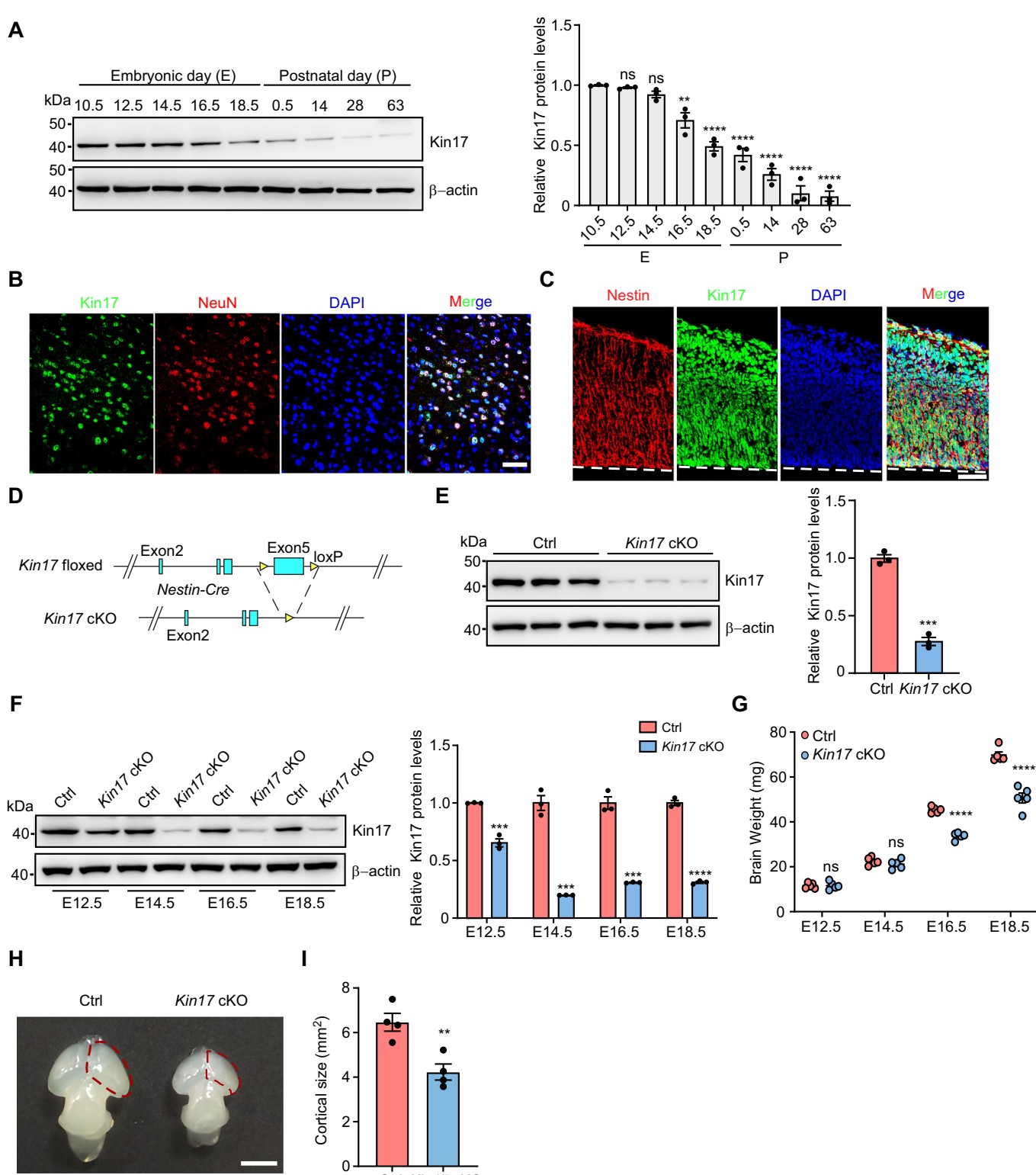

**A** Embryonic day (E)   Postnatal day (P)

**B** Kin17   NeuN   DAPI   Merge

**C** Nestin   Kin17   DAPI   Merge

**D** Kin17 floxed   Nestin-Cre   Kin17 cKO

**E** Ctrl   Kin17 cKO

**F** Ctrl   Kin17 cKO

**G**

**H** Ctrl   Kin17 cKO

**I**

downregulated in Kin17 cKO cortices compared to controls (Fig. 3A). Gene Ontology (GO) analysis of the downregulated genes revealed significant enrichment for terms such as "central nervous system development" and "forebrain development", while upregulated genes were linked to "immune system process" (Figs. 3B and EV3A). Given

the pivotal role of NPC proliferation and differentiation in brain development (Homem et al, 2015), we focused on the downregulated genes, which were enriched for terms including "neurogenesis", "neural precursor cell proliferation", and "cell differentiation". These gene expression patterns were visualized in a heatmap and further

**Figure 1. Depletion of Kin17 in neural progenitor cells leads to brain developmental atrophy.**

(A) Kin17 expression in the brain of WT mice at embryonic day (E) 10.5, 12.5 ($P > 0.9999$), 14.5 ($P > 0.9999$), 16.5 ($P = 0.0017$), and 18.5 ($P < 0.0001$), and at postnatal day (P) 0.5 ($P < 0.0001$), 14 ($P < 0.0001$), 28 ($P < 0.0001$), and 63 ($P < 0.0001$), by immunoblotting and densitometry. β-actin was included as a loading control ($n = 3$ per group). (B) Representative immunofluorescence images of Kin17 (green), NeuN (red), and DAPI (blue) in 3-month-old WT brain. Scale bars, 50 μm. (C) Representative immunofluorescence images of Kin17 (green), Nestin (red), and DAPI (blue) in the cortex of WT mice at E14.5. Scale bars, 50 μm. (D) Generation of Kin17 conditional knockout mice. Schematic of the genomic target sites in *Kin17* locus. (E) The expression of Kin17 in control and *Kin17* cKO mice cortex at E14.5, by immunoblotting and densitometry ($P = 0.0001$). β-actin was included as a loading control ($n = 3$ per group). (F) Kin17 expression in the cortical brain of control and *Kin17* cKO mice at E12.5 ($P = 0.0006$), E14.5 ($P = 0.0002$), E16.5 ($P = 0.0002$), and E18.5 ($P < 0.0001$), by immunoblotting and densitometry. β-actin was included as a loading control ($n = 3$ per group). (G) Brain weight of control and *Kin17* cKO mice at E12.5 ($P > 0.9999$), E14.5 ($P > 0.9999$), E16.5 ($P < 0.0001$), and E18.5 ($P < 0.0001$) ($n = 5$ per group). (H) Morphological comparison of control and Kin17 cKO brains. Red dashed box indicates the cerebral cortex. Scale bar, 2 mm. (I) Quantification of cortical area for the indicated cerebral cortex in (H) ($P = 0.0058$) ($n = 4$ per group). Statistical analysis was performed using one-way ANOVA with Bonferroni correction (A), two-way ANOVA with Bonferroni correction (G) or two-tailed Student's *t* test (E, F, I). Error bars denote the SEM. **$P < 0.01$; ***$P < 0.001$; ****$P < 0.0001$; ns not significant.

validated by qPCR analysis (Fig. EV3B,C). Collectively, these findings suggest that Kin17 depletion in NPCs leads to brain atrophy by impairing both the proliferation and differentiation of NPCs.

Cortical neurons are derived from NPCs, predominantly comprising RGs and IPs (Tang et al, 2019). RGs and IPs are characterized by distinct markers: Pax6 for RGs and Tbr2 for IPs (Lv et al, 2019; Manuel et al, 2015). Immunostaining at E16.5 revealed a significant reduction in Pax6- and Tbr2-positive progenitor numbers in the cortices of *Kin17* cKO mice compared to controls (Fig. 3C–E), indicating impaired NPC proliferation due to Kin17 depletion. These differences were not observed at E14.5 (Fig. EV3D–F), suggesting that the effects of Kin17 loss manifest after this stage.

Phosphorylated histone 3 (PH3), a mitotic marker specifically expressed in NPCs (Qiao et al, 2018). Immunostaining showed a marked decrease in PH3-positive cells in Kin17 cKO cortices at E16.5 compared to controls (Fig. 3F,G), further confirming reduced NPC proliferation following Kin17 depletion. No differences were detected at E14.5 (Fig. EV3G,H).

Bromodeoxyuridine (BrdU), a thymidine analog, can be integrated into newly synthesized DNA to monitor cell proliferation rate (Wojtowicz and Kee, 2006). To directly assess proliferation, we performed BrdU labeling, wherein pregnant mice were intraperitoneally injected with BrdU and embryos were analyzed either 1 or 24 h later. At one hour post-injection, BrdU-positive cells represent actively proliferating cells, while at 24 h, BrdU-positive cells represent both proliferating and differentiated cells. We found that Kin17 cKO brains showed significantly fewer BrdU-positive cells at 1 h (Fig. 3H,I), indicating reduced proliferation.

Ki67 serves as a known proliferation marker, with expression initiating at the onset of late G1 and ceasing upon cell cycle exit (Uxa et al, 2021). BrdU-positive/Ki67-positive (BrdU+/Ki67+) cells reflect those that remain actively in DNA replication, while BrdU-positive/Ki67-negative (BrdU+/Ki67−) cells reflect those that have exited the cell cycle and begun differentiation. The number of both BrdU+/Ki67+ and BrdU+/Ki67− cells was significantly reduced in Kin17 cKO cortices (Fig. 3J–L), suggesting impaired proliferation and differentiation. We further evaluated neuronal differentiation using Tuj1, a marker protein of postmitotic neurons. Immunostaining at E16.5 revealed a substantial reduction in Tuj1-positive neurons in Kin17 cKO cortices compared to controls (Fig. 3M), consistent with impaired neurogenesis. These findings collectively demonstrate that Kin17 is essential for maintaining proper NPC proliferation and differentiation during brain development, and that its loss disrupts cortical lamination and leads to brain atrophy.

## Kin17 promotes proliferation and differentiation in neurospheres

Neurospheres are in vitro cultures that model the behavior of NPCs (da Silva Siqueira et al, 2021). To assess the role of Kin17 in NPC proliferation and differentiation, we cultured neurospheres derived from the cortices of control and *Kin17* cKO mice. Immunoblotting analysis confirmed the absence of Kin17 expression in cKO neurospheres (Fig. 4A). Notably, neurospheres derived from *Kin17* cKO mice exhibited significantly smaller diameters compared to those from control mice (Fig. 4B,C), indicating impaired proliferation following Kin17 depletion. Consistently, MTT assays revealed a markedly slower growth rate in Kin17 cKO neurospheres (Fig. 4D), indicating that Kin17 is required for normal NPC proliferation in vitro.

We further evaluated proliferation by conducting immunostaining for Ki67, a marker of mitotic activity (Singec et al, 2006), in combination with Nestin, a progenitor marker. The number of Nestin+/Ki67+ cells was significantly reduced in Kin17 cKO neurospheres compared to controls (Fig. 4E,F), reinforcing the conclusion that Kin17 loss impairs NPC proliferation. Similarly, BrdU incorporation assays revealed significantly fewer BrdU-positive cells in Kin17 cKO neurospheres than in controls (Fig. 4G,H), providing further evidence of reduced proliferative capacity.

Since NPC proliferation and differentiation are tightly linked processes during brain development (Zahr et al, 2019), we next assessed the impact of Kin17 on neuronal differentiation. Differentiation was induced by withdrawing growth factors from the neurosphere culture medium and successful induction was confirmed by robust Tuj1 expression in control neurospheres (Fig. 4I). In contrast, significantly fewer Tuj1-positive cells were observed in differentiated Kin17 cKO neurospheres compared to control neurospheres (Fig. 4I,J), indicating impaired neuronal differentiation. These findings collectively demonstrate that Kin17 is essential for both NPC proliferation and differentiation in neurospheres, supporting its critical role in establishing the laminated structure of the developing cortex.

## Kin17 promotes ribosome biogenesis by enhancing rDNA transcription

To investigate how Kin17 regulates NPC proliferation and differentiation, we conducted RNA-seq on neurospheres derived control and *Kin17* cKO mice. By comparing the RNA-seq data from neurosphere and cortical tissue, we identified 1,401 DEGs that were

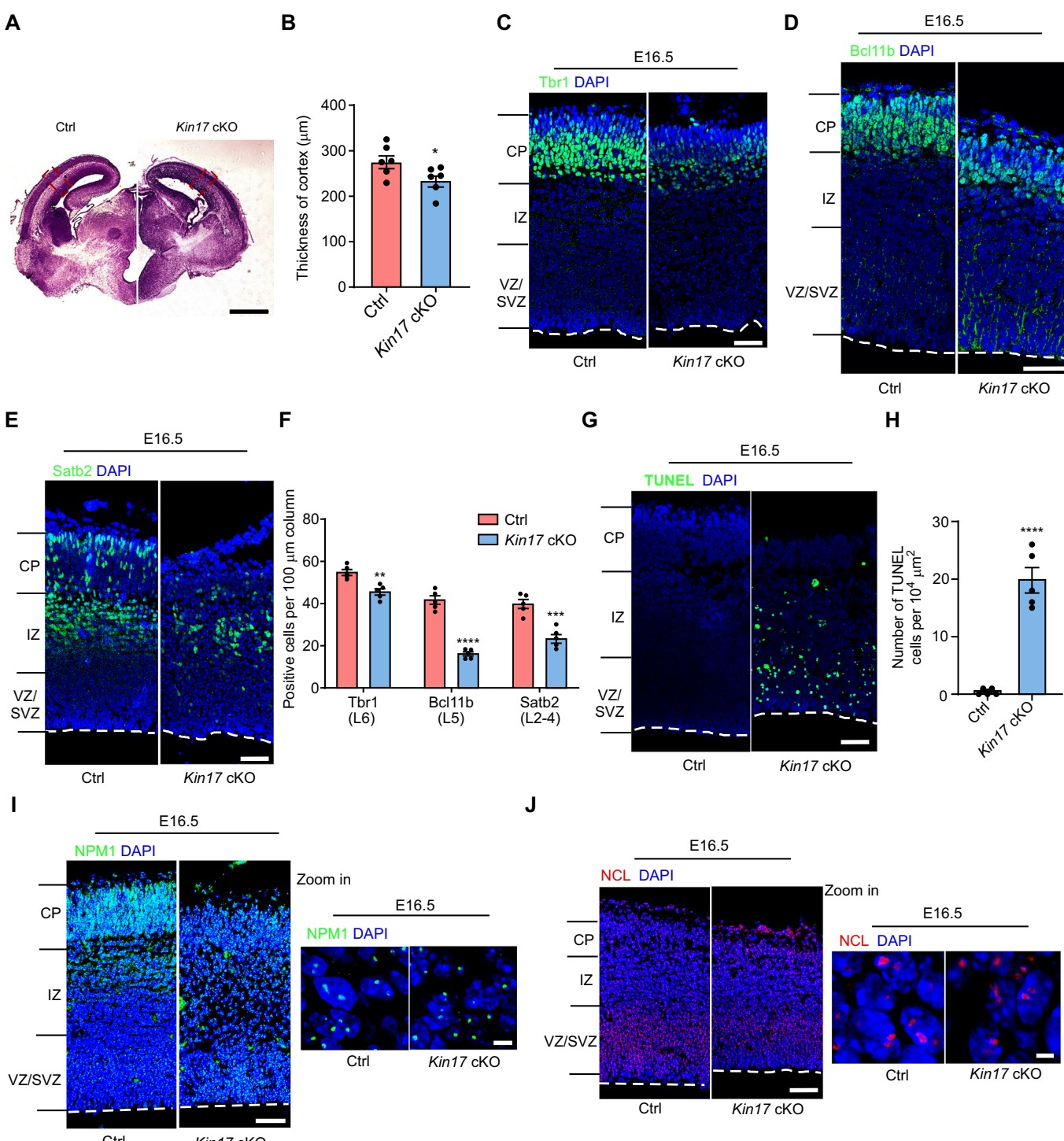

commonly altered in Kin17 cKO versus control samples (Fig. 5A). GO analysis of these shared DEGs revealed enrichment in terms such as "structural constituent of ribosome", "cytoplasmic translation", and "cytosolic ribosome" (Fig. EV4A). By performed immunostaining, we detected a colocalization of Kin17 and NCL, indicating a physical localization of Kin17 in the nucleolus (Fig. EV4B). These lines of evidence indicate that Kin17 plays a critical role in regulating ribosome structure and function.

Given that ribosomal RNA (rRNAs) are essential for ribosome formation and that rRNA synthesis is a rate-limiting step in ribosome biogenesis (Falcon et al, 2022), we measured the levels of 45S precursor rRNA and mature 28S, 18S, and 5.8S rRNAs in neurospheres. We observed a significant decrease in all these rRNAs in Kin17 cKO neurospheres compared to controls (Fig. 5B), indicating that Kin17 suppresses rRNA production. A similar reduction in 45S pre-rRNA was observed in E16.5 Kin17 cKO, but

Figure 2. Depletion of Kin17 in NPCs leads to defective cortex lamination.

(A) Nissl staining for control and Kin17 cKO brain at E16.5. Scale bar, 500 μm. (B) Quantification of the cortical thickness for the indicated red dashed box in (A) (P = 0.0468) (n = 6 for each group). (C–F) Representative immunofluorescence images of Tbr1 (green) and DAPI (blue) (C), of Bcl11b (green) and DAPI (blue) (D), of Satb2 (green) and DAPI (blue) (E), in control and Kin17 cKO brain at E16.5. The white border indicates the apical surface of VZ. Scale bar, 50 μm. (F) Quantification of cells positive for the indicated markers in both control and Kin17 cKO brains (Tbr1, P = 0.0017, Bcl11b, P < 0.0001, Satb2, P = 0.0006) (n = 5 per group). (G) Representative images of TUNEL staining (green) and DAPI (blue) in the cortex of control and Kin17 cKO mice at E16.5. Scale bar, 50 μm. (H) Quantification of TUNEL-positive cells in the cortices of control and Kin17 cKO mice (P < 0.0001) (n = 5 per group). (I) Representative immunofluorescence images of NPM1 (green) and DAPI (blue) in the cerebral cortex of control and Kin17 cKO mice at E16.5. Scale bar, 50 μm (left) and 5 μm (right). (J) Representative immunofluorescence images of NCL (red) and DAPI (blue) in the cerebral cortex of control and Kin17 cKO mice at E16.5. Scale bar, 50 μm (left) and 3 μm (right). Statistical analysis was performed using two-tailed Student's t test (B, F, H). Error bars denote the SEM. *P < 0.05; **P < 0.01; ***P < 0.001; ****P < 0.0001.

not in E14.5 Kin17 cKO cortices, when compared to their respective controls (Figs. 5C and EV4C), supporting the onset of defective cortical lamination at E16.5. To directly assess rDNA transcription, we treated neurosphere cultures with 4-thiouridine (4sU), a natural uridine derivative incorporated into RNA transcripts, followed by qPCR to detect 4sU-incorporated rRNAs to indicate nascent transcripts. Levels of newly synthesized 45S pre-rRNA were markedly reduced in Kin17 cKO neurospheres relative to control neurospheres (Fig. 5D), confirming that Kin17 promotes rDNA transcription. Moreover, we examined intermediate rRNA processing products using northern blot analysis. In addition to a reduction in 45S rRNA, we observed a significant decrease in intermediate rRNA products in the cortex of Kin17 cKO mice, including 34S, 32S, and 12S (Fig. EV4D). However, the ratio of each intermediate product to the 45S precursor remained comparable between Kin17 cKO and control cortices (Fig. EV4E), suggesting that Kin17 has no impact on rRNA processing.

Our RNA-seq data revealed that a total of 31 Rpl genes and 29 Rps genes were significantly downregulated in Kin17 cKO neurospheres compared to controls (Fig. EV4F). Among ribosomal proteins, large subunit Rpl12, Rpl4, and small subunit Rps3, Rps14 exhibited the most pronounced reductions in Kin17 cKO neurospheres compared to controls. We selected these representative ribosomal proteins for further validation and confirmed their downregulation in Kin17 cKO neurospheres (Fig. 5E). These findings further support that loss of Kin17 disrupts ribosome biogenesis.

To assess the impact of Kin17 on ribosome formation, we conducted polysome profiling to separate and analyze the small subunits (40S), large subunits (60S), monosomes (80S), and polysomes (mRNA-ribosome complexes with multiple ribosomes). We observed a significant reduction in 40S, 60S, 80S monosomal, and polysomes in Kin17 KD 293T cells compared to control 293T cells (Fig. 5F,G). Of note, the ratios of 40S and 60S to 80S did not differ between control and Kin17 KD 293T cells (Fig. 5H), indicating that Kin17 has no impact on monosome assembly. Similarly, the monosome-to-polysome ratio was comparable between control and Kin17 KD cells (Fig. 5H), suggesting that polysomal formation is also unaffected. These results support a role of Kin17 in ribosome biogenesis, specifically influencing the overall production of ribosomal subunits without impairing their assembly into functional ribosomes.

To evaluate ribosome function, we performed a puromycin incorporation assay and found a significant decrease in puromycin-incorporated nascent proteins in Kin17 cKO neurospheres compared to control neurospheres (Fig. 5I), demonstrating global protein synthesis impairment. In summary, these results indicate that Kin17 facilitates ribosome biogenesis and function by promoting rRNA transcription and ribosome biogenesis.

## Kin17 promotes rDNA transcription by sequential recruitment of NCL and RNA polymerase I

As a known DNA-binding protein, we hypothesized that Kin17 might directly regulate rDNA transcription. Chromatin immunoprecipitation followed by qPCR analysis (ChIP-qPCR) using an anti-Kin17 antibody showed that Kin17 binds to the promoter region of 45S rDNA. We included UBF, a known positive control that binds to the promoter of rDNA (O'Sullivan et al, 2002; Sanij and Hannan, 2009) as well as Kin17 mutant lacking the zinc finger as a negative control in this analysis (Figs. 6A and EV5A–C). We further investigated the global distribution of Kin17 along the rDNA locus, with primers targeting multiple distinct regions of the rDNA, including the enhancer, promoter, 5' external transcribed spacer (5'ETS), 18S, 28S, 5.8S, intergenic spacer (IGS), and transcription termination site (TTS) (Fig. EV5D). Our results revealed that Kin17 was predominantly enriched at the rDNA promoter (Fig. 6B). As a positive control, we included the global distribution of UBF across the rDNA locus, which is known to bind to the enhancer, promoter, and 5'ETS regions (Fig. 6B).

To validate this interaction, we generated a luciferase reporter construct containing a human rDNA promoter sequence (pHrD-IRES2-luc) and transfected it into control or Kin17 KD 293T cells (Ghoshal et al, 2004). Reporter assays revealed significantly decreased luciferase activity in Kin17 KD cells compared to controls (Fig. 6C), indicating that Kin17 enhances rDNA promoter activity.

Actively transcribed rDNA is typically unmethylated, and ChIP-Chop analysis allows for the discrimination of rDNA methylation status (Santoro, 2014). Using HpaII restriction enzyme digestion, followed by PCR detection with specific primers (Fig. 6D), we observed no significant difference in rDNA methylation between control and Kin17 cKO neurospheres (Fig. EV5E), suggesting that Kin17 has no impact on the global methylation state of rDNA. By conducting chromatin IP using an anti-Kin17 antibody, followed by PCR detection, we found that the proportion of methylated rDNA bound by Kin17 was significantly lower than that present in the input (Fig. 6E), indicating that Kin17 is preferentially enriched at unmethylated rDNA loci. The methylation state of UBF-bound rDNA, which is known to be predominantly unmethylated, was assessed and included as a positive control (Fig. 6E). These results support the conclusion that Kin17 preferentially associates with unmethylated rDNA to facilitate rDNA transcription.

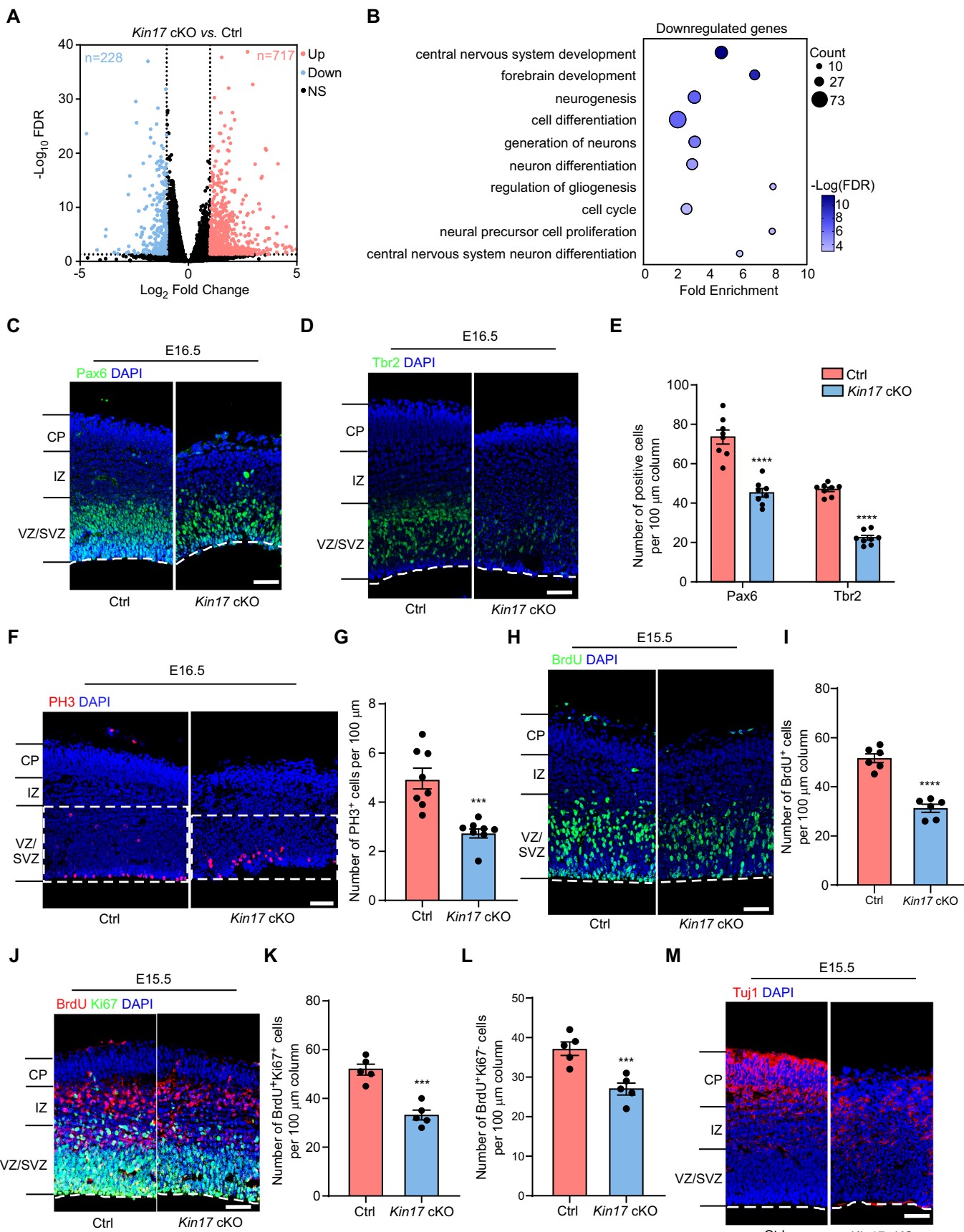

◄  **Figure 3.  Deficiency of Kin17 impairs the proliferation and differentiation of NPCs.**

(A) A volcano plot showing the differentially expressed genes (DEGs) in cerebral cortex of *Kin17* cKO versus control mice at E16.5 ($n = 3$ for each genotype). The blue dots represent downregulated DEGs, while the red dots represent upregulated DEGs. (B) A bubble plot illustrating GO enrichment analysis of downregulated genes in the cerebral cortex of *Kin17* cKO versus control mice at E16.5. (C) Representative immunofluorescence images of Pax6 (green) and DAPI (blue) in the cerebral cortex of *Kin17* cKO versus control mice at E16.5. The white border indicates the apical surface of VZ. Scale bar, 50 μm. (D) Representative immunofluorescence images of Tbr2 (green) and DAPI (blue) in the cerebral cortex of *Kin17* cKO versus control mice at E16.5. The white border indicates the apical surface of VZ. Scale bar, 50 μm. (E) Quantitative analysis for Pax6 ($P = 0.000009$) or Tbr2 ($P < 0.000001$) positive cells in control and Kin17 cKO cortices at E16.5 ($n = 8$ per group). (F) Representative immunofluorescence images of PH3 (red) and DAPI (blue) in the cerebral cortex of control and *Kin17* cKO mice at E16.5. The white border indicates the VZ/SVZ. Scale bars, 50 μm. (G) Quantitative analysis for PH3-positive cells in control and *Kin17* cKO mice ($P = 0.0003$) ($n = 8$ per group). (H, I) Pregnant mice at E15.5 were intraperitoneally injected with BrdU ($n = 6$ per group). (H) Representative immunofluorescence images of BrdU (green) and DAPI (blue) in the fetal cerebral cortex, 1 h after the injection. The white border indicates the apical surface of VZ. Scale bars, 50 μm. (I) Quantitative analysis for BrdU-positive cells ($P < 0.0001$) ($n = 6$ per group). (J–L) Pregnant mice at E14.5 were intraperitoneally injected with BrdU ($n = 5$ per group). (J) Representative immunofluorescence images of BrdU (red), Ki67 (green), and DAPI (blue) in the fetal cerebral cortex, 24 h after the injection. The white border indicates the apical surface of VZ. Scale bars, 50 μm. (K) Quantitative analysis for BrdU-positive and Ki67-positive cells ($P < 0.0003$) ($n = 5$ per group). (L) Quantitative analysis for BrdU-positive and Ki67-negative cells ($P = 0.001$) ($n = 5$ per group). (M) Representative immunofluorescence images of Tuj1 (red) and DAPI (blue) in the cerebral cortex of control and Kin17 cKO cortices at E16.5. The white border indicates the apical surface of VZ. Scale bar, 50 μm. Statistical analysis was performed using two-tailed *t* test (E, G, I, K, L). Error bars denote the SEM. ***$P < 0.001$; ****$P < 0.0001$.

Methylation for Kin17 regulates its chromatin-binding capacity, whereby the unmethylated form preferentially associates with chromatin, while methylated form tends to dissociate from chromatin (Cloutier et al, 2014). This modification is known to be catalyzed by PRMT7 (Gaspar et al, 2021). We generated a Kin17 mutant (R36K), in which the arginine at position 36 was substituted with lysine, tagging with an HA epitope on the N-terminus. This mutant is a previously characterized PRMT7-binding-deficient variant (Gaspar et al, 2021). N2a cells were transfected with either HA-tagged WT Kin17 or R36K mutant, along with Flag tagged Prmt7 for expression. The interaction of Kin17 and PRMT7 was assessed by co-immunoprecipitation (co-IP) using an anti-Flag antibody, followed by immunoblotting with an anti-HA antibody. We demonstrated a robust interaction of PRMT7 and WT Kin17, whereas the R36K mutant showed a significantly reduced association (Fig. EV5F), indicating that methylation at the R36 residue is critical for Kin17-PRMT7 interaction. To assess the functional consequence of this mutant, we conducted ChIP-qPCR analyses using an anti-HA antibody to evaluate rDNA occupancy. Our results demonstrate that the R36K mutant showed significantly higher rDNA binding compared to WT Kin17 in the presence of exogenous PRMT7 (Fig. EV5G), suggesting that unmethylated Kin17, but not its methylation form, is competent for rDNA binding. We further determined the impact of Kin17 R36K mutant on 45S pre-rRNA level and our results revealed that Kin17 R36K mutant showed a significantly higher 45S level compared to WT Kin17 in the presence of exogenous PRMT7 (Fig. EV5H), suggesting that unmethylated Kin17 preferentially binds to rDNA and promotes its transcription.

To explore the underlying mechanism of Kin17 and rDNA binding, we performed immunoprecipitation (IP) followed by mass spectrometry (MS) to identify Kin17-associated proteins. Nucleolin (NCL), a known regulator of rDNA transcription (Durut and Sáez-Vásquez, 2015), emerged as a top candidate (Fig. EV6A; Table EV3). Co-immunostaining of NCL and Sox2, a NPC marker, revealed colocalization of the two proteins, indicating that NCL indeed is expressed in NPCs (Fig. EV6B). Co-IP confirmed the interaction between endogenous Kin17 and NCL in both cultured neurospheres and cortical tissues (Figs. 6F and EV6C). This interaction was further validated in 293T cells co-expressing HA-Kin17 and Flag-NCL (Figs. 6G and EV6D). To determine whether the interaction of Kin17 and NCL is RNA dependent, we have

performed co-IP in the presence or absence of RNase A. We detected a similar extent of Kin17-NCL association in the presence and absence of RNase A (Fig. EV6E), suggesting that the interaction of Kin17 and NCL is independent of RNA.

To investigate how Kin17 interacts with NCL, we generated two truncation mutants of Kin17: Kin17-M1 (lacking the N-terminal DNA binding domain) and Kin17-M2 (lacking the C-terminal RNA-binding domain), both tagged with GST (Fig. 6H). Co-transfection with Flag-NCL revealed that Kin17-M1 still interacted with NCL, whereas Kin17-M2 did not (Fig. 6I), indicating that Kin17 interacts with NCL through its RNA-binding domain. Similarly, truncation fragments of NCL showed that only the C-terminal fragment containing RNA-binding domains (NCL-M2), and not the N-terminal intrinsically disordered region (NCL-M1), interacted with GST-Kin17 (Fig. 6H,J). These findings suggest that Kin17 and NCL interact via their respective C-terminal RNA-binding domains.

NCL has been recognized for its role in regulating rDNA transcription (Li et al, 2018). Consistent with this, we observed that NCL KD significantly decreased the levels of 45S pre-rRNA and mature rRNAs in neurospheres (Fig. EV6F). rDNA reporter assays also showed diminished luciferase activity in NCL KD cells (Fig. EV6G). Notably, dual knockdown of NCL and Kin17 did not further reduce 45S pre-rRNA levels compared individual knockdowns (Fig. 6K), suggesting that NCL-Kin17 interaction is crucial for rDNA transcription.

Given that RNA polymerase I (Pol I) is responsible for transcribing rDNA in eukaryotes and the chromatin loading of Polr1a, the catalytic subunit of the Pol I complex, reflects the enrichment of the Pol I complex on chromatin and its transcriptional activity (Günzl et al, 2003; Xu et al, 2016b). NCL facilitates rDNA transcription by ensuring proper Pol I loading onto rDNA (Cong et al, 2012). Co-IP confirmed interactions between NCL and Polr1a (Fig. EV6H), as well as between Kin17 and Polr1a. Importantly, the protein levels of both NCL and Polr1a remained unchanged in *Kin17* cKO neurospheres (Fig. EV6I), suggesting that Kin17 regulates rDNA transcription via recruitment rather than regulation of protein expression.

ChIP-qPCR analysis revealed significantly reduced occupancy of both Pol I and NCL at the rDNA promoter in Kin17 KD N2a cells compared to control cells (Fig. 6L,M), indicating that their recruitment to rDNA depends on Kin17. These findings suggest

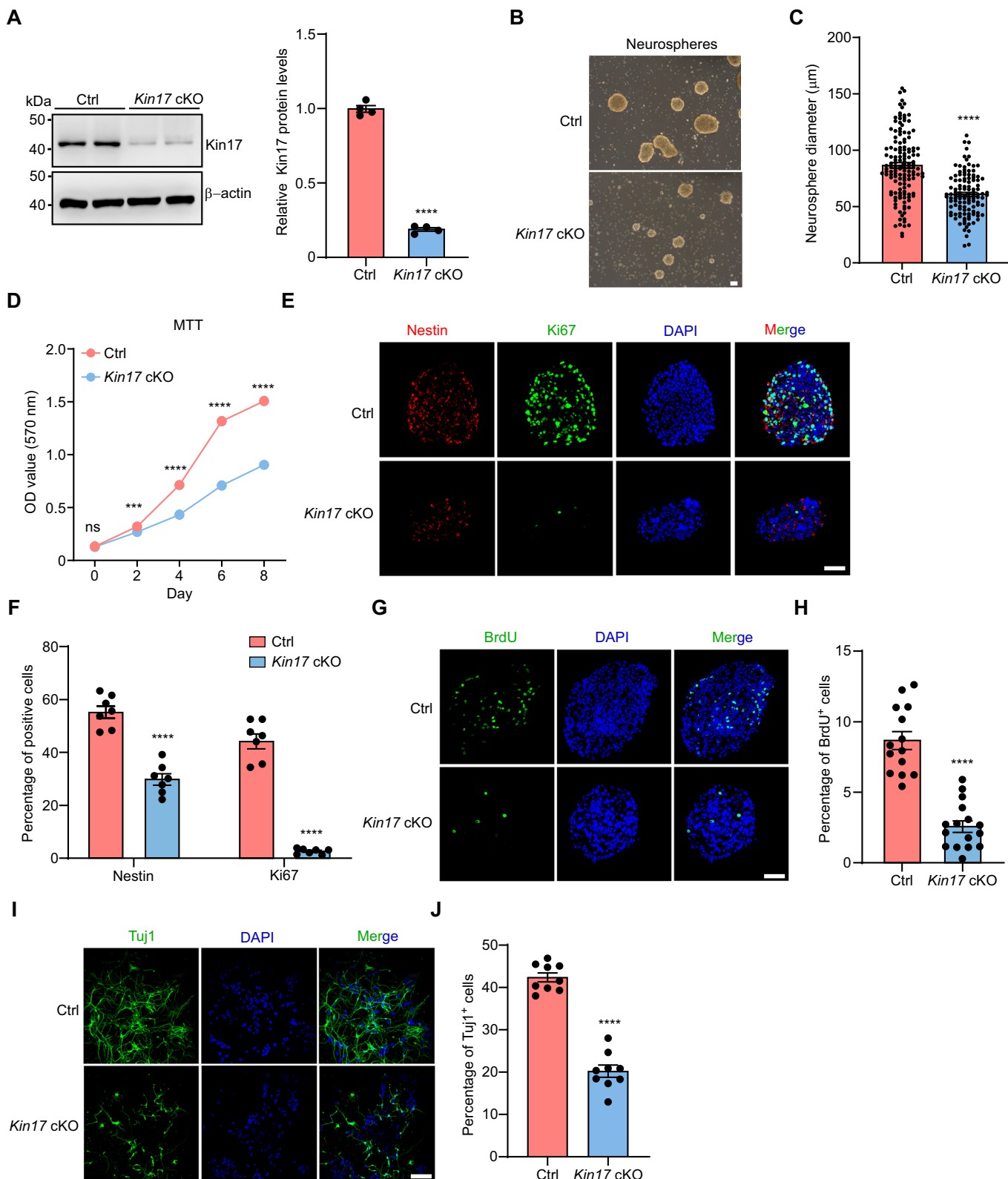

**Figure 4.  Kin17 promotes neurosphere proliferation and differentiation.**

(A) Kin17 expression in control and Kin17 cKO neurospheres, by immunoblotting and densitometric analysis ($P < 0.0001$) ($n = 3$). β-actin was included as a loading control. (B) Representative phase-contrast images of primary cultured neurospheres derived from control and *Kin17* cKO mice at E14.5. Scale bar, 50 µm. (C) Diameter measurements of neurosphere derived from control ($n = 140$) and *Kin17* cKO ($n = 115$) mice at E14.5 and in vitro cultured for 5 days ($P < 0.0001$). (D) Cell proliferation rate of control and Kin17 cKO neurospheres at the indicated culture days (day 0, $P = 0.9949$; day 2, $P = 0.0007$; day 4, $P < 0.0001$; day 6, $P < 0.0001$; day 8, $P < 0.0001$) ($n = 6$ per group). (E) Representative immunostaining images of Nestin (red), Ki67 (green), and DAPI (blue) in control and Kin17 cKO neurospheres. (F) Quantification analysis for Nestin positive cells ($P < 0.0001$) or Ki67-positive cells ($P < 0.0001$) in (E) ($n = 7$ per group). (G, H) Control and Kin17 cKO neurospheres were incubated with BrdU for 1 h. (G) Representative immunostaining images of BrdU (green) and DAPI (blue). Scale bars, 50 µm. (H) Quantification analysis for BrdU-positive cells in control ($n = 14$) and Kin17 cKO neurospheres ($n = 16$) ($P < 0.0001$). (I, J) Neuronal differentiation was induced in control and Kin17 cKO neurospheres for 7 days. (I) Representative immunostaining images of Tuj1 (green) and DAPI (blue). Scale bar, 50 µm. (J) Quantification analysis of Tuj1-positive neurons ($P < 0.0001$) ($n = 9$ per group). Statistical analysis was performed using two-way ANOVA with Bonferroni correction (D) or two-tailed Student's $t$ test (A, C, F, H, J). Error bars denote the SEM. \*\*\*$P < 0.001$; \*\*\*$P < 0.0001$; ns not significant.

the formation of a Kin17-NCL-Pol I complex involved in rDNA transcription regulation.

Considering that neither NCL nor Pol I possess intrinsic DNA binding sequences, we hypothesized that the DNA-binding protein Kin17 facilitates Pol I-mediated rDNA transcription by recruiting NCL. Supporting this model, ChIP-qPCR analysis showed a significant reduction in Polr1a occupancy at the rDNA promoter in NCL KD N2a cells, while Kin17 occupancy remained unchanged (Fig. 6L,N). Furthermore, simultaneous knockdown of Kin17 and NCL did not lead to an additive decrease in Polr1a loading (Fig. 6L), reinforcing the idea that Kin17 acts upstream of NCL to facilitate Pol I recruitment. Together, these results demonstrate that Kin17 promotes rDNA transcription by sequentially recruiting NCL and Pol I to the rDNA promoter.

## Discussion

Embryonic brain development is a complex process involving the proliferation and differentiation of neuronal stem/progenitor cells (NPCs), neuronal migration, and the establishment of polarity (Dong et al, 2021; Hakanen et al, 2019; Nakagawa et al, 2020). During early development, NPCs located in the ventricular and subventricular zone undergo self-renewal divisions to expand the progenitor pool (Koo et al, 2023). As development progresses, these cells undergo asymmetric divisions and ultimately differentiate into neurons, contributing to the formation of the a six-layered cortical laminar structure (Agirman et al, 2017). In this study, we demonstrate that Kin17 is vital for brain development in mice by maintaining normal proliferation and differentiation of NPCs. Mechanistically, we demonstrate that Kin17 promotes rDNA transcription by recruiting NCL, thereby facilitating the loading of the Pol I complex onto the rDNA promoter and subsequently enhancing ribosomal biogenesis.

Kin17 is a highly conserved gene that is ubiquitously expressed in mammals (le Maire et al, 2006). Kin17 has been recognized as a component of multiprotein complex involved in DNA replication, transcription, and cell cycle regulation (Miccoli et al, 2005). Kin17 expression is upregulated in response to DNA damage, where it contributes to the maintenance of fidelity and genome stability (Biard et al, 2002; Biard et al, 1997; Masson et al, 2003). In *C. elegans*, Kin17 has been identified as a splicing factor essential for maintaining 5' splice site identity during spliceosome assembly (Suzuki et al, 2022). A similar role of Kin17 has been proposed in humans, based on evidence of Kin17's interaction with components of the spliceosome (Gaspar et al, 2021; Rappsilber et al, 2002). In our analysis of mouse embryos, we found that Kin17 is abundantly expressed during the embryonic stage. Notably, homozygous deletion of Kin17 results in embryonic lethality, underscoring its critical role in early development. In contrast, conditional knockout of *Kin17* (cKO) specifically in NPCs allows for survival through developmental, but leads to impaired cortex lamination, highlighting its specific importance in neuro-development processes.

Asymmetric cell division for NPCs enables both self-renewal and the generation of differentiated neurons (Royall et al, 2023). This process must be tightly regulated to maintain the proper balance between the progenitor pool and newly formed neurons (Homem et al, 2015). As a DNA- and RNA-binding protein, Kin17 was recently identified as a novel regulatory of asymmetric cell division in *Drosophila* neuroblasts (Connell et al, 2024). In this context, Kin17 guides the proper localization of Miranda, which ensures the proper segregation of the fate determinant proteins during asymmetric cell division (Connell et al, 2024). In our study, we demonstrate that Kin17 is essential for maintaining both the proliferation and neuronal differentiation of NPCs. These findings agree with previous studies showing that Kin17 is necessary for the proliferation of thyroid or cervical cancer (Jiang et al, 2021; Zhang et al, 2017).

The cerebral cortex is characterized by a six layered organization, where newly generated neurons from NPCs migrate into the cortical plate, ultimately forming the laminated structure of cortex (Greig et al, 2013). We observed that the absence of Kin17 results in reduced neuronal numbers across multiple cortical layers, including deep layer (V and layer VI), upper layer II/III, and layer IV, thereby impairing the structural integrity of the cerebral cortex.

Ribosomes, the central machinery for protein translation, consist of a small (40S) and a large (60S) subunit. The small subunit contains one rRNA (18S) and 33 proteins, while the large subunit comprises three rRNAs (28S, 5.8S, and 5S) and 47 proteins (Dörner et al, 2023). The 45S pre-rRNA, transcribed by RNA polymerase I, is processed into the mature rRNAs (5.8S, 18S, and 28S) through a series of coordinated endonucleolytic and exonucleolytic cleavages (Henras et al, 2015). It has been reported that rRNA expression plays a pivotal role in controlling NPC proliferation during development (Falcon et al, 2022; Nakagawa et al, 2020). In our study, we show that the loss of Kin17 in NPCs leads to impaired brain development, including reduced cortical size and thickness. Mechanistically, we demonstrate that Kin17

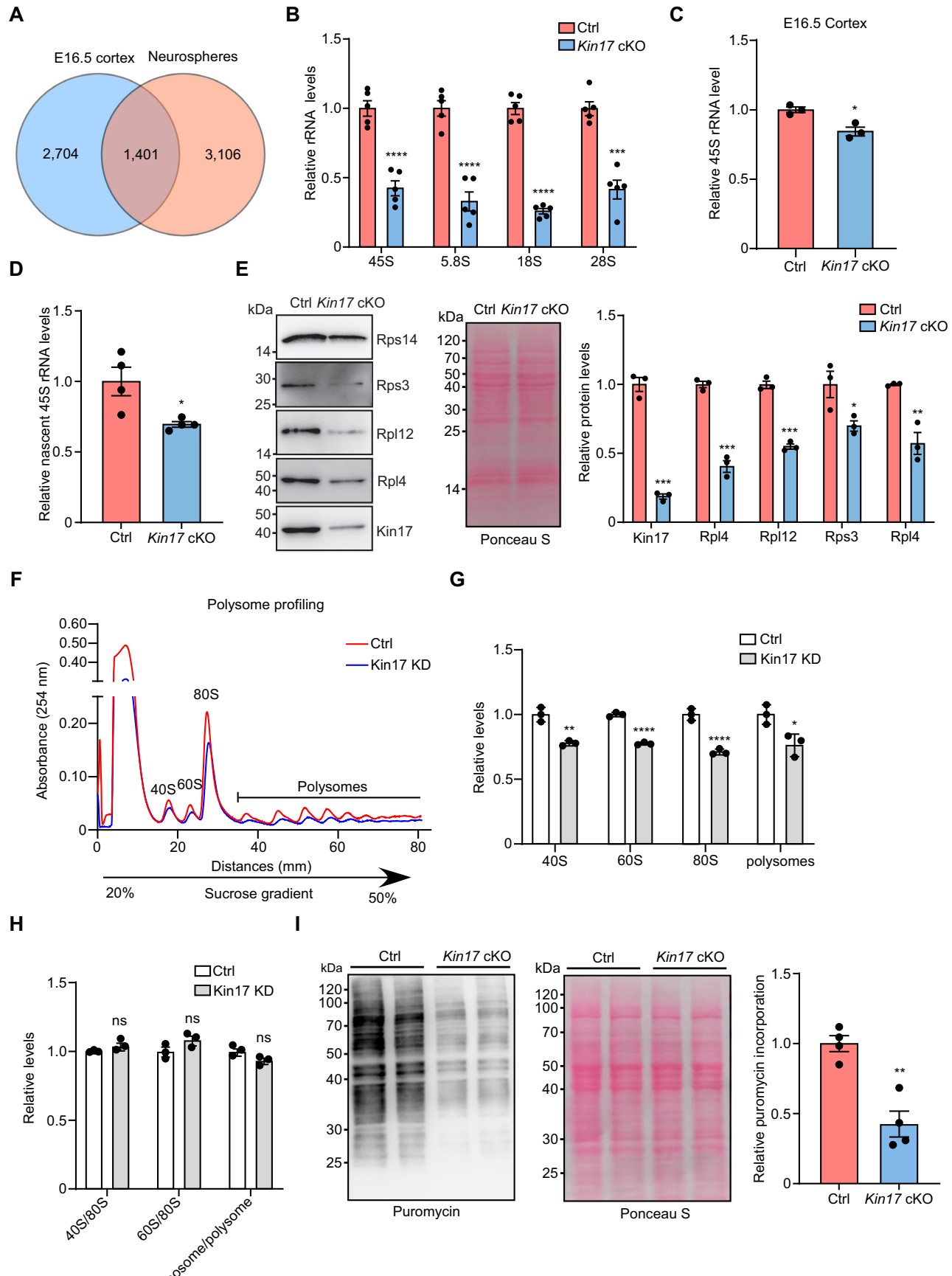

◄ **Figure 5. Kin17 promotes ribosome biogenesis by facilitating rDNA transcription.**

(A) Venn diagram showing the number of DEGs in Kin17 cKO versus control neurosphere, Kin17 cKO versus control E16.5 cortices, and the overlapping DEGs between the two datasets. (B) The levels of rRNA (45S, 28S, 18S, and 5.8S) in control and Kin17 cKO neurospheres, by qPCR (45S, $P < 0.0001$; 28S, $P < 0.0001$; 18S, $P < 0.0001$; 5.8S, $P = 0.0001$). Data were normalized against U1 snRNA ($n = 5$ per group). (C) 45S rRNA level was determined in control and Kin17 cKO cortices at E16.5, by qPCR analysis ($P = 0.0142$) ($n = 3$). (D) Nascent 45S pre-rRNA levels in control and Kin17 cKO neurospheres following 1-h 4sU labeling, by qPCR ($P = 0.02$). Data were normalized against U1 snRNA ($n = 4$ per group). (E) Levels of large ribosomal subunit protein (Rpl4, Rpl12) and small ribosomal subunit protein (Rps3, Rps14) in control and Kin17 cKO neurospheres, by immunoblotting and densitometric analyses (Kin17, $P = 0.000111$; Rpl4, $P = 0.000271$; Rpl12, $P = 0.000165$; Rps3, $P = 0.045249$; Rps14, $P = 0.005937$). Ponceau S staining ensured equal protein loading ($n = 3$ per group). (F) Cytoplasmic ribosome distribution in control (red) and Kin17 KD (blue) 293T cells. (G) Quantification of area under curve (AUC) for 40S, 60S, 80S, and polysomes in control and Kin17 KD 293T cells (40S, $P = 0.001961$, 60S, $P = 0.000045$, 80S, $P = 0.000992$, polysome, $P = 0.026363$) ($n = 3$). (H) Ratios of 40S to 80S ($P = 0.249865$), 60S to 80S ($P = 0.176167$), and monosome to polysome ($P = 0.132058$) were determined in control and Kin17 KD 293 T cells ($n = 3$). (I) Control and Kin17 cKO neurospheres were fed with puromycin for 1 h. Puromycin incorporated proteins were determined by immunoblotting and densitometric analyses ($P = 0.0019$). Ponceau S staining ensured equal protein loading ($n = 4$ per group). Statistical analysis was performed using two-tailed Student's $t$ test (B–E, G–I). Error bars denote the SEM. *$P < 0.05$; **$P < 0.01$; ****$P < 0.0001$; ns not significant.

deficiency suppresses the transcription of 45S pre-rRNA and the production of mature rRNAs (5.8S rRNA, 18S rRNA, and 28S rRNA), thereby reducing ribosome biogenesis and impairing translational function of ribosome.

Ribosome biogenesis requires precise coordination between the production of rRNA and ribosomal protein. Rrn3 (also known as TIF1-A in vertebrates) protein is known to bind RNA polymerase I (RNAPI) and activates rDNA transcription by facilitating its recruitment to the 35S rDNA gene promoters. In addition, Rrn3 simultaneously regulates rRNA and ribosomal protein gene (RPG) transcription following inhibition of target of rapamycin complex 1 (TORC1) kinase, when ribosome biogenesis is strongly downregulated. This trans-effect on RNPII is specific to RPGs, providing strong evidence for a regulatory crosstalk between RNAPI and RNAPII that coordinates rRNA and RPG transcription (Laferté et al, 2006).

RPG transcription is also governed by transcriptional activator Ifh1, which cooperates with its promoter-bound partner Fhl1. Under optimal growth condition, Ifh1 binds to multiple RPG promoters to enhance their expression. Upon TORC1 inhibition or nutrient deprivation, Ifh1 is rapidly released from RP promoters. This release is dependent on Utp22, which sequesters Ifh1 activity and prevents its binding to RPG promoters. Importantly, Utp22-mediated inhibition of Ifh1 activity is inversely proportional to RNAPI activity, supporting a mechanistic link between rRNA production and RPG transcription (Albert et al, 2016). Additionally, Ifh1 promoter binding is coordinated with RNA polymerase I activity upon prolonged TORC1 inhibition to maintain a balance of RP and rRNA production (Schawalder et al, 2004). Moreover, Ifh1 is a component of CURI complex, which includes several proteins involved in pre-rRNA processing, suggesting a role of Ifh1 in coupling RPG transcription and pre-rRNA processing (Rudra et al, 2007). Taken together, these lines of evidence demonstrate that rRNA production and RPG transcription are tightly linked to maintaining a balanced ribosome biogenesis.

NCL is a nucleolar protein known to play a critical role in the regulation of rRNA biosynthesis and processing (Okuwaki et al, 2021). In this study, we demonstrate that NCL KD reduced the levels of both 45S pre-rRNA and mature rRNAs in neurospheres, consistent with previous findings in embryonic stem cells and neurons (Li et al, 2018; Percharde et al, 2018). Prior studies have reported that NCL KD leads to the accumulation of Pol I at the transcription start site, accompanied by reduced binding of

upstream binding factor (UBF) to both the coding and promoter regions of rDNA(Cong et al, 2012). Our findings further demonstrate that Kin17 recruits NCL and facilitates rDNA transcription through its interaction with the Pol I complex. This suggests that Kin17-NCL interaction may serve as a scaffold molecule to recruit Pol I complex onto the rDNA promoter, thereby enhancing rDNA transcription. Of note, NCL binding to the rDNA promoter is not entirely dependent on Kin17, implying that other factors may also contribute to its recruitment. Indeed, recent studies have shown that non-coding rRNAs can act as molecular scaffolds to guide NCL binding to rDNA loci (Percharde et al, 2018).

Previous study reported an association between Kin17 and the SSU processome by employing proximity-dependent biotin identification coupled to mass spectrometry (BioID-MS), supporting a potential role for Kin17 in rRNA processing (Gaspar et al, 2021). Our study experimentally demonstrates that Kin17 binds to rDNA and promotes its transcription, without affecting rRNA processing. We believe this discrepancy may arise from differences in the experimental systems used: while this report utilized human cell lines such as 293 and HeLa, our study employed cultured mouse neurospheres to investigate the Kin17-associated proteins. It is also important to note that the BioID approach used in the referenced study, which relies on proximity-dependent biotinylation, may capture not only true interacting proteins but also those in close proximity, potentially leading to false positives—especially when identifying highly abundant proteins (Sears et al, 2019).

Mice homozygous for a knockout of Polr1a, a core subunit of Pol I, exhibit embryonic lethality prior to implantation. Conditional ablation of Polr1a in forebrain progenitors results in a hypoplastic telencephalon and a marked reduction in telencephalon size (Smallwood et al, 2023). Similarly, depletion of Polr1a or Polr1c in the neuroepithelium and neural crest cells (NCCs) leads to NCC apoptosis and severe craniofacial abnormalities (Falcon et al, 2022). Notably, NCL depletion produces similar defects in embryonic craniofacial development (Dash and Trainor, 2022). While there is currently no direct evidence linking Polr1a or NCL to cortical development, these prior findings suggest that their loss could plausibly contribute to cortical atrophy.

Taken together, our study uncovers a critical role for Kin17 in neurogenesis during embryonic development through regulating ribosome biogenesis and function, subsequently NPC proliferation and differentiation.

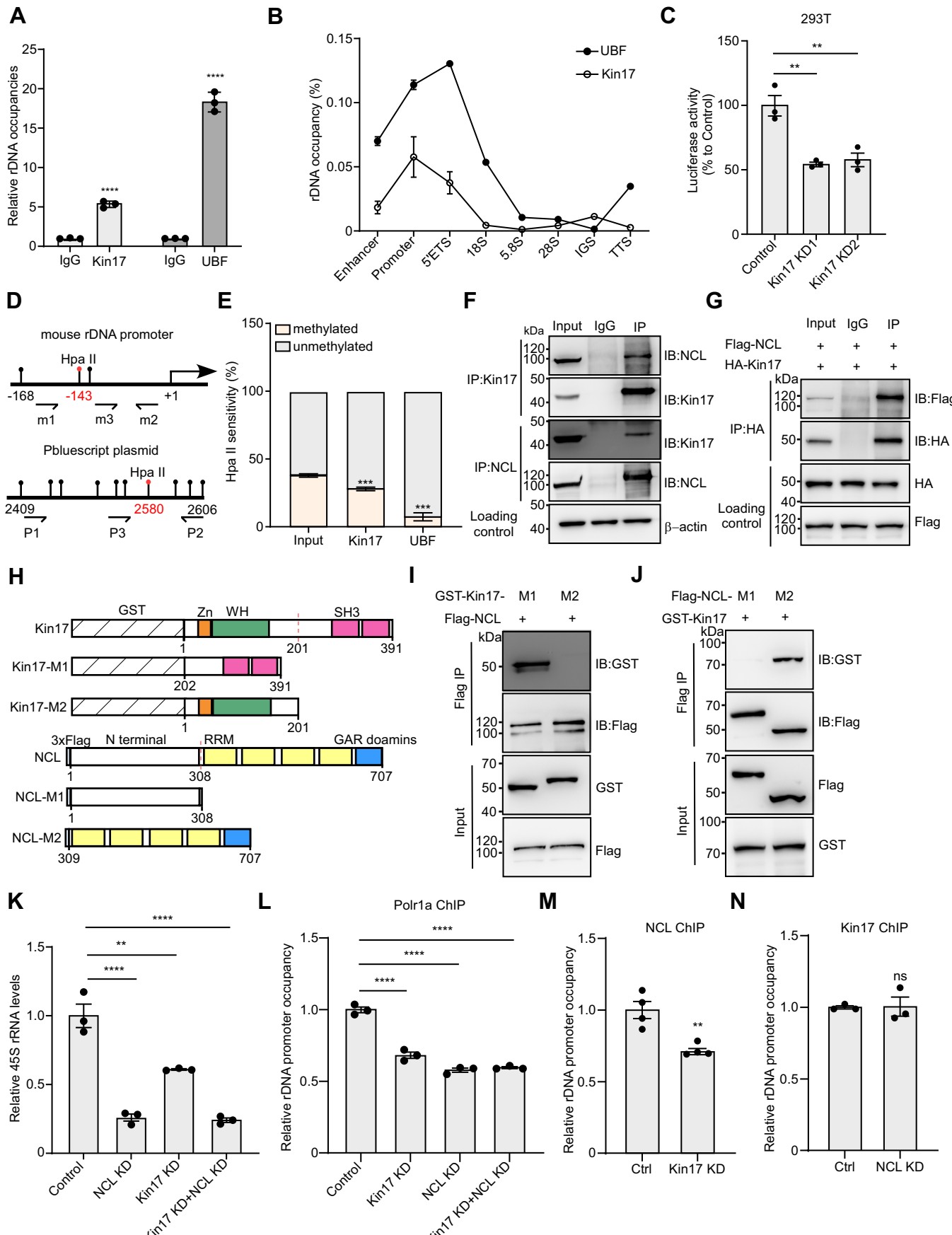

◄

**Figure 6. Kin17 promotes rDNA transcription through sequential recruitment of NCL and RNA polymerase I.**

(A) Neurospheres were subjected to ChIP assays using an anti-Kin17 antibody, followed by qPCR-based detection of rDNA (Kin17, $P = 0.000053$; UBF, $P = 0.000017$) ($n = 3$). UBF was included as positive control. (B) The global distribution pattern of Kin17 and UBF along the rDNA locus was determined by ChIP-qPCR. Kin17- or UBF-bound rDNA was quantified relative to input rDNA ($n = 3$). (C) rDNA promoter reporter plasmid was transfected into control or Kin17 KD 293T cells. Luciferase activity was determined by Dual-Luciferase reporter assay (Kin17 KD1, $P = 0.0024$; Kin17 KD2, $P = 0.0035$) ($n = 3$ per group). (D) Schematic illustrating the experimental strategy for measuring CpG methylation at rDNA promoter regions. Mouse rDNA was digested with the HpaII enzyme, followed by PCR detection using the indicated primers. Digestion efficiency was assessed by PCR amplification of Pbluescript sequences using the corresponding primers. (E) Kin17-associated rDNA was obtained by chromatin IP using an anti-Kin17 antibody. The relative abundance of methylated rDNA was quantified relative to input. As a positive control, UBF associated rDNA was isolated via chromatin IP using an anti-UBF antibody and the corresponding methylation status was determined. The distributions of methylated (orange bars) and unmethylated (grey bars) rDNA copies were plotted accordingly (Kin17, $P = 0.0004$; UBF, $P = 0.0006$) ($n = 3$). (F) The interaction of Kin17 and NCL in neurospheres, determined by co-IP with an anti-Kin17 antibody, followed by immunoblotting detection with an anti-NCL antibody, and reciprocally by co-IP with an anti-NCL antibody followed by immunoblotting detection with an anti-Kin17 antibody ($n = 3$). (G) The interaction of exogenous HA-Kin17 and Flag-NCL in 293T cells co-transfected with HA-Kin17 and Flag-NCL, determined by co-IP with an anti-HA antibody and immunoblotting detection with an anti-Flag antibody ($n = 3$). (H) Generation of Kin17 or NCL truncation fragments. (I) Full-length Flag-NCL and each of truncated GST-Kin17 plasmids were transfected into 293T cells for expression, followed by co-IP with an anti-Flag antibody and immunoblotting with an anti-GST antibody ($n = 3$). (J) Full-length GST-Kin17 and each of truncated Flag-NCL plasmids were transfected into 293T cells for expression, followed by co-IP with an anti-Flag antibody and immunoblotting with an anti-GST antibody ($n = 3$). (K) 45S pre-rRNA level in control N2a cells, N2a cells with NCL KD ($P < 0.0001$), N2a cells with Kin17 KD ($P = 0.001$), and N2a cells with both NCL and Kin17 KD ($P < 0.0001$), by qPCR ($n = 3$ per group). (L) Relative Polr1a enrichment at the promoter of 45S rDNA in control N2a cells, N2a cells with NCL KD ($P < 0.0001$), N2a cells with Kin17 KD ($P < 0.0001$), and N2a cells with both NCL and Kin17 KD ($P < 0.0001$), by ChIP-qPCR ($n = 3$). (M) Relative NCL enrichment on the promoter of 45S rDNA in control and Kin17 KD N2a cells, by ChIP-qPCR analysis ($P = 0.004262$) ($n = 3$). (N) Relative Kin17 enrichment on the promoter of 45S rDNA, in control and NCL KD N2a cells, by ChIP-qPCR analysis ($P = 0.801241$) ($n = 3$). For this and subsequent figures, KD represents knockdown. Statistical analysis was performed using one-way ANOVA (C, K, L) with Bonferroni correction or two-tailed Student's $t$ test (A, E, M, N). Error bars denote the SEM. **$P < 0.01$, ***$P < 0.001$; ****$P < 0.0001$; ns not significant.

# Methods

### Reagents and tools table

| Reagent/resource | Reference or source | Identifier or catalog number |
|---|---|---|
| **Experimental models** | | |
| C57BL/6 (*M. musculus*) | Shanghai Model Organisms Center | SM-001 |
| *Kin17^{flox/+}* (*M. musculus*) | Biocytogen Pharmaceuticals | |
| *Kin17^{+/−}* (*M. musculus*) | | |
| *Nestin-Cre* (*M. musculus*) | Jackson Lab | Cat #003771 |
| 293T cells (*Homo sapiens*) | ATCC | CRL-3216 |
| N2a cells (*M. musculus*) | ATCC | CCL-131 |
| Neurospheres (*M. musculus*) | Lin et al, 2020 | |
| **Recombinant DNA** | | |
| pLVX-CMV | Clontech | Cat #632565 |
| pcDNA3.1(+) | Thermo Fisher Scientific | Cat #V79020 |
| pGreenPruo | SBI | Cat #SI505A-1 |
| pmirGlo | Promega | Cat #E1330 |
| pYIC | Addgene | Cat #18673 |
| pGL3 | Promega | Cat #E1751 |
| Pbluescript (pBS) | Addgene | Cat #212205 |
| Kin17 R36K | This study | |
| **Antibodies** | | |
| Rabbit polyclonal anti-Bcl11b | Proteintech | Cat# 55414-1-AP |
| Mouse monoclonal anti-BrdU | BD Pharmingen | Cat# 555627 |
| Mouse monoclonal anti-Flag | Enogene | Cat# E12-001 |
| Rabbit polyclonal anti-Flag | Enogene | Cat# E12-021 |

| Reagent/resource | Reference or source | Identifier or catalog number |
|---|---|---|
| Mouse monoclonal anti-HA | Enogene | Cat# E12-003 |
| Rabbit polyclonal anti-HA | Proteintech | Cat# 51064-2-AP |
| Rabbit polyclonal anti-Ki67 | Abcam | Cat# ab15580 |
| Rabbit polyclonal anti-Kin17 | Proteintech | Cat# 12313-1-AP |
| Mouse monoclonal anti-Kin17 | Santa Cruz | Cat# sc-32769 |
| Rabbit polyclonal anti-NCL | Proteintech | Cat# 10556-1-AP |
| Mouse monoclonal anti-Nestin | Cell Signaling Technology | Cat# 4760 |
| Rabbit polyclonal anti-Pax6 | Proteintech | Cat# 12323-1-AP |
| Rabbit polyclonal anti-PH3 | Millipore | Cat# 06-570 |
| Rabbit polyclonal anti-Polr1a | Proteintech | Cat# 20595-1-AP |
| Mouse monoclonal anti-Puromycin | Millipore | Cat# MABE343 |
| Rabbit polyclonal anti-Rpl4 | Proteintech | Cat# 11302-1-AP |
| Rabbit polyclonal anti-Rps14 | Proteintech | Cat# 16683-1-AP |
| Rabbit polyclonal anti-Rpl12 | Proteintech | Cat# 14536-1-AP |
| Rabbit polyclonal anti-Rps3 | Proteintech | Cat# 11990-1-AP |
| Rabbit polyclonal anti-Satb2 | Abcam | Cat# ab92446 |
| Mouse monoclonal anti-Sox2 | Proteintech | Cat# 66411-1-Ig |
| Rabbit polyclonal anti-Tbr1 | Proteintech | Cat# 20932-1-AP |
| Rabbit monoclonal anti-Tbr2 | Abcam | Cat# ab183991 |
| Mouse monoclonal anti-Tuj1 | Proteintech | Cat# 66375-1-Ig |
| Mouse monoclonal anti-β-actin | Affnity | Cat# T0022 |

| Reagent/resource | Reference or source | Identifier or catalog number |
|---|---|---|
| Goat anti-Mouse IgG Secondary Antibody, Alexa Fluor 488 | Invitrogen | Cat# A-11001 |
| Goat anti-Mouse IgG Secondary Antibody, Alexa Fluor 594 | Invitrogen | Cat# A-11005 |
| Goat Anti-Mouse IgG-HRP | Elabscience | Cat# E-AB-1003 |
| Goat Anti-Rabbit IgG-HRP | Elabscience | Cat# E-AB-1001 |
| Goat anti-Rabbit IgG Secondary Antibody, Alexa Fluor 594 | Invitrogen | Cat# A-11012 |
| Goat anti-Rabbit IgG Secondary Antibody, Alexa Fluor 488 | Invitrogen | Cat# A-11034 |
| **Oligonucleotides and other sequence-based reagents** | | |
| Genotyping primers | This study | Table EV1 |
| Cloning primers | This study | Table EV1 |
| shRNA sequences | This study | Table EV1 |
| ChIP-qPCR primers | This study | Table EV2 |
| qPCR primers | This study | Table EV2 |
| Northern blot probes | Li et al, 2018; Shcherbik et al, 2010 | Table EV2 |
| ChIP-Chop primers | Santoro, 2014 | Table EV2 |
| **Chemicals, enzymes and other reagents** | | |
| BrdU | Topscience | Cat# 59-14-3 |
| 4-thiouridine (4sU) | Sigma-Aldrich | Cat #T4509 |
| Ponceau S | Sangon Biotech | Cat# A610437 |
| Cresyl Violet Acetate | Sangon Biotech | Cat# A606312 |
| T4 DNA Ligase | Vazyme | Cat# C301 |
| HiScript Reverse Transcriptase | Vazyme | Cat# R201 |
| AceQ qPCR SYBR Green Master Mix | Vazyme | Cat# Q121 |
| DMEM | Gibco | Cat #12100061 |
| Accutase | Gibco | Cat #A1110501 |
| Neurobasal | Gibco | Cat #21103049 |
| B27 | Gibco | Cat #17504044 |
| bFGF | MCE | Cat #HY-P7066 |
| EGF | MCE | Cat #HY-P7067 |
| poly-D-lysine | Sigma-Aldrich | Cat #P7886) |
| PEI | Sigma-Aldrich | Cat #408727 |
| EB | Thermo Fisher Scientific, | Cat #15585011 |
| DTT | GoldBio | Cat #27565-41-9 |
| Protease inhibitor cocktail | Targetmol | Cat #C0001 |
| BCA Protein Assay kit | Thermo Fisher Scientific | Cat #23227 |
| nitrocellulose membranes | Pall Corporation | Cat #27182369 |
| enhanced chemiluminescence (ECL) substrates | Thermo Fisher Scientific | Cat #32132 |
| biotin-HPDP | Thermo Fisher Scientific | Cat #21341 |

| Reagent/resource | Reference or source | Identifier or catalog number |
|---|---|---|
| PCR purification kit | Sangon | Cat #B518191 |
| DAPI | Sigma-Aldrich | Cat #D9542 |
| GlutaMAX | Gibco | Cat #35050079 |
| FBS | Biological Industries | Cat #04-001-1ACS |
| Dual Luciferase Reporter Gene Assay Kit | Yeasen | Cat #11402ES60 |
| **Software** | | |
| DAVID | https://david.ncifcrf.gov/ | |
| Gene Ontology | http://geneontology.org/ | |
| Heatmapper | http://www.heatmapper.ca/ | |
| VENNY 2.1.0 | Venny 2.1.0 (csic.es) | |
| GraphPad Prism v10.1.2 | https://www.graphpad.com/ | |
| NCBI | ncbi.nlm.nih.gov | |
| Primer 3 | Primer3 Input | |
| **Other** | | |
| protein A/G magnetic beads | MCE | Cat #HY-K0202 |
| Illumina Novaseq platform | Annoroad Gene Technology | |
| streptavidin beads | Thermo Fisher Scientific | Cat #62210) |
| DH5α | Vazyme | Cat #C502-02 |

## Animals

C57BL/6 WT mice were purchased from the Shanghai Model Organisms Center (Shanghai, China). *Nestin-Cre* transgenic mice were originally obtained from the Jackson Laboratory. *Kin17* heterozygous knockout (*Kin17*^{+/−}) mice were generated using CRISPR technology to delete exon 5 of the *Kin17* gene. *Kin17*^{flox/flox} mice were generated by flanking exon 5 with two lox P sites. Both mouse lines were developed by Biocytogen Pharmaceuticals (Beijing, China). *Kin17*^{flox/flox} mice were crossed with *Nestin-Cre* transgenic mice to produce *Kin17*^{flox/flox}/*Nestin-Cre* (*Kin17* cKO) offspring. All mice were housed under specific-pathogen-free (SPF) conditions on a 12-h dark/light cycle at $21 \pm 1\,°C$ and 55–60% humidity, with ad libitum access to water and standard chow. All animal experiments were approved by the Animal Ethics Committee of the University of Science and Technology of China (USTCACUC23120123040).

## Cell culture

293T cells or Neuro 2a (N2a) cells were cultured under standard conditions in DMEM supplemented with 10% FBS and 1% penicillin/streptomycin (P/S). Cells were maintained in a humidified incubator at $37\,°C$ with 5% $CO_2$.

## Neurosphere culture and in vitro differentiation

Neurosphere culture was prepared as previously published (Lin et al, 2020). Briefly, E14.5 mouse cortices were dissected and enzymatically dissociated using 0.05% Trypsin-EDTA at 37 °C for 2 min. The digestion was quenched with DMEM containing 10% FBS. The cell suspension was centrifuged at $500 \times g$ for 5 min at room temperature, and the resulting pellet was resuspended in Neurobasal supplemented with 2% B27, 2 mM GlutaMAX, 1% P/S, 20 ng/mL basic fibroblast growth factor (bFGF), and 20 ng/mL epidermal growth factor (EGF). Neural progenitor cells were plated at a density of $2 \times 10^5$ cells per well in Ultra-Low Attachment 6-well plates and incubated for 5–7 days at 37 °C in 5% $CO_2$.

For differentiation assays, neurospheres were dissociated into single cells using Accutase for 8 min at 37 °C, followed by trituration in differentiation medium consisting of Neurobasal medium with 2% B27, 2 mM GlutaMAX, and 1% P/S. Dissociated cells were then seeded at a density of $1 \times 10^5$ cells per well on glass coverslip pre-coated with poly-D-lysine and Matrigel, and cultured for an additional 7 days.

## Plasmid construction

The full-length coding sequence of mouse *Kin17* was amplified by PCR from mouse cDNA and subcloned into a modified pLVX-CMV vector, in which the ZsGreen element was removed. N-terminal 3 × HA tag was fused to Kin17. Similarly, the coding sequence of mouse NCL was PCR-amplified and subcloned into the pLVX-CMV vector with 3 × Flag tag on the N-terminus. Coding sequence of Prmt7 was amplified by PCR from mouse cDNA and subcloned into pLVX-CMV vector, with 3 × Flag tag constructed on its N-terminus. Kin17 truncation fragment sequences were PCR-amplified from mouse cDNA and inserted into the pcDNA3.1(+) vector, with an N-terminal GST tag. Kin17 mutant lacking the zinc finger was PCR-amplified from mouse cDNA and inserted into pcDNA3.1(+) vector with 3 × Flag tag on the N-terminus. Kin17 mutation (R36K) was generated by site-directed mutagenesis. NCL truncation fragments were subcloned into the pLVX-CMV vector with N-terminal 3 × Flag tag. Kin17 and NCL shRNAs sequences were synthesized by Songon (Shanghai, China) and cloned into the pGreenPruo lentiviral vector. All cloning primers and shRNA sequences are listed in Table EV1.

## RNA extraction, reverse transcription, and qPCR

Total RNA was extracted from mouse cortices or cultured neurospheres using TRIzol reagent, following the manufacturer's instructions. cDNA was synthesized using the HighScript II First-Strand Synthesis System according to the manufacturer's protocol. qPCR was performed using the LightCycler 96 system with AceQ qPCR SYBR Green Master Mix. The real-time value for each sample was averaged and compared using the Ct method, where the amount of target RNA ($2^{-\Delta\Delta Ct}$) was normalized to an endogenous reference ($\Delta Ct$). PCR efficiency and linearity were assessed using serially diluted cDNA (0.1 ng, 0.01 ng, 0.001 ng, and 0.0001 ng), as previously described (Shehata et al, 2019). Primer sequences used for qPCR are listed in Table EV2.

## Genotyping

Genomic DNA was extracted from mouse tail biopsies by incubation in 50 mM NaOH at 95 °C for 30 min, followed by neutralization with 100 mM Tris-HCl (pH 7.5). The resulting DNA was used as a template for PCR amplification using Taq DNA polymerase (Nova Biomedical), according to the manufacturer's instructions. PCR products were resolved by agarose gel electrophoresis and visualized using ethidium bromide (EB). Genotyping primer sequences are listed in Table EV1.

## Protein extraction and western blotting

Cells or brain tissues were lysed in RIPA buffer containing 50 mM Tris-HCl pH 7.5, 150 mM NaCl, 1% sodium deoxycholate, 0.5% NP-40, 0.1% SDS and 1% protease inhibitor cocktail. Lysates were sonicated and centrifuged at $20,000 \times g$ for 10 min at 4 °C, and the supernatants were collected. Protein concentrations were determined using the BCA Protein Assay kit. Equal amounts of protein were separated by SDS-PAGE and transferred to 0.45-µm nitrocellulose membranes. Membranes were blocked with 5% non-fat milk in TBST (TBS, 0.1% Tween-20) for 1 h at room temperature and incubated overnight at 4 °C with primary antibody diluted in blocking buffer. Membranes were washed with TBST for five times, 5 min for each time, then incubated with HRP-conjugated secondary antibody in 5% milk for 1 h at room temperature. Immunoreactive bands were detected using enhanced chemiluminescence (ECL) substrates and visualized using Chemi-Scope (CLiNX). Band intensities were quantified using ImageJ software (NIH). Antibodies used for western blotting are listed in the Reagents and Tools Table.

## Lentivirus production and neurosphere transduction

Lentiviral plasmids, including the expression construct and packaging (pHR'8.2deltaR) and enveloping (pCMV-VSV-G) plasmids were co-transfected into 293T cells at ~80% confluency using PEI at a 3:2:1 ratio. Virus-containing medium was harvested 48 h post-transfection and concentrated by ultracentrifugation. The viral pellet was resuspended in PBS, and viral titers were determined by a qPCR-based approach. For transduction, 2 µL of control shRNA virus ($1.5 \times 10^9$ TU/mL), or 4 µL of NCL shRNA1 or NCL shRNA2 virus (each at $7.5 \times 10^8$ TU/mL), was added to neurosphere cultures in six-well plates.

## Dual-Luciferase reporter assay

To construct pGL3-hRluc (Renilla luciferase) vector, hRluc and its SV40 promoter were PCR-amplified from the pmirGlo vector and cloned into the pGL3-basic vector. The pHrD-IRES2-luc reporter was generated based on a previously published with minor modifications (Ghoshal et al, 2004). Briefly, IRES2 was PCR-amplified from the pYIC vector and inserted into the pGL3-basic vector to create pGL3-IRES2. The human rDNA promoter sequence (from −410 to +314 relative to the transcription start site) was PCR-amplified from 293T genomic DNA and cloned upstream of the IRES2 element in pGL3-IRES2 vector. Cloning primers are listed in Table EV1.

The activity of rDNA promoter was determined by Dual Luciferase Reporter Gene Assay Kit according to the manufacturer's instructions. Briefly, pHrD-IRES2-luc and pGL3-hRluc plasmid were co-transfected into 293T cells at a 10:1 ratio. After 48 h, cells were lysed, and luminescence was measured using a Microplate Reader (Molecular Devices, SpectraMax i3). Firefly luciferase activity was normalized to Renilla luciferase signals.

## Immunofluorescence staining

Embryonic brains were dissected, rinsed in PBS, and fixed in 4% PFA overnight at 4 °C. Fixed tissues were cryoprotected in 20% and 30% sucrose solutions, then embedded in O.C.T. compound, and sectioned at 20 μm thickness using a microtome. Neurospheres were collected by centrifugation at $500 \times g$ for 3 min at 4 °C, washed with cold PBS twice, fixed in 4% PFA for 15 min, and rinsed with PBS for three times. Dissociated neurons were treated similarly.

Cryosections or cultured cells were permeabilized and blocked in a solution of 2% BSA containing 0.4% Triton X-100 for 45 min at room temperature. After incubation with primary antibodies overnight at 4 °C, brain sections or cells were washed with PBS, followed by incubation with Alexa Fluor 488-conjugated Goat anti-rabbit secondary antibody or Alexa Fluor 594-conjugated Goat anti-mouse secondary antibody for 1 h at room temperature. DNA was counterstained with DAPI. Fluorescence images were acquired using a Leica confocal laser microscope equipped with DFC 365 FX Digital Camera.

## BrdU labeling

Pulse-labeling with bromodeoxyuridine (BrdU) was performed by intraperitoneal injection of BrdU to pregnant mice (50 mg/kg). Embryonic brains were fixed in 4% PFA, cryosectioned, treatment of 2 M HCl for 30 min at room temperature, followed by neutralization in 0.1 M sodium borate buffer (pH 8.5). BrdU incorporation was detected using an anti-BrdU antibody by immunofluorescence staining.

## Nissl staining

Brain sections (20 μm) were washed three times with PBS for 5 min each. Brian sections were then submerged in 0.1% cresyl violet acetate solution for 3 min. After staining, sections were rinsed with distilled water to remove the residual staining solution, dehydrated sequentially in 95% ethanol and 100% ethanol for 2 min each, cleared in xylene for 2 min, and mounted with neutral resin.

## RNA metabolic labeling

RNA metabolic labeling was performed as previously described (El-Brolosy et al, 2019; Sun and Chen, 2018), with modifications. Neurosphere cultures were fed with 200 μM 4-thiouridine (4sU) for 1 h, followed by total RNA extraction using Trizol. A total of 100 μg of RNA was incubated with biotin-HPDP to biotinylate the newly transcribed RNAs. Excess biotin-HPDP was removed using Phase Lock Gel, and biotinylated RNA was captured using streptavidin beads. 4sU-labeled RNA was eluted with 100 mM DTT and analyzed by qPCR.

## Puromycin incorporation assay (SUnSET assay)

The SUnSET assay was performed according to previously published protocol (Schmidt et al, 2009), with modifications. Briefly, neurospheres were treated with 10 μg/mL puromycin for 1 h. Puromycin-incorporated proteins were detected by western blotting using an anti-puromycin antibody. Ponceau S staining was used as a loading control.

## Chromatin immunoprecipitation (ChIP)

ChIP was carried out as previously described (Li et al, 2021), with minor modifications. Neurospheres or N2a cells were crosslinked with 0.75% formaldehyde for 10 min, followed by quenching with 150 mM glycine for 10 min. Cross-linked cells were harvested by centrifugation at $2000 \times g$ for 5 min at 4 °C and washed twice with cold PBS. Cells were lysed in ice-cold lysis buffer (50 mM Tris pH 8.0, 140 mM NaCl, 1 mM EDTA, 10% glycerol, 0.5% NP-40, 0.25% Triton X-100) for 20 min. Cell lysates were subjected to centrifugation at $2000 \times g$ for 5 min at 4 °C, and the resulting pellet was resuspended in an ice-cold lysis buffer (10 mM Tris-HCl pH 8.0, 1 mM EDTA, 10% glycerol, 0.5% NP-40 and 0.25% Triton X-100). Chromatin was sheared to an average fragment size of 200–1000 bp using Covaris M220 (65 s, 50 W, 20% duty cycle). Sheared chromatin was incubated overnight at 4 °C with protein A/G magnetic beads pre-bound to the appropriate antibodies. After extensive washing, beads were resuspended in an elution buffer (1% SDS, 100 mM NaHCO$_3$), followed by treatment with RNase A and proteinase K. Crosslinks was reversed, and DNA was released and purified using a PCR purification kit. The purified DNA was analyzed by qPCR. ChIP-qPCR primer sequences are listed in Table EV2.

## The ChIP-Chop assay

ChIP-Chop assay was conducted as previously described (Santoro, 2014). ChIP and input samples (20 μL each) were digested by adding 2 μL of HpaII, 3 μL 10× NEBuffer 1, 2 μL of 1 ng/μL Pbluescript (pBS) plasmid, and 3 μL of water, followed by incubation at 37 °C for 2 h. After heat inactivation for 10 min, 70 μL of water was added to each sample, and DNA was purified using the SanPrep Column PCR Product Purification Kit. To assess the methylation status of the CpG residues at position −143 in the mouse rDNA promoter, 2 μL of HpaII-digested DNA from both ChIP and input samples were amplified using primers m1 and m2, which flank the CpG site. The total amount of rDNA promoter sequence in each sample was quantified by amplifying a region from −105 to −1 (lacking HpaII sites) using primers m3 and m2. The efficiency of HpaII digestion was assessed by PCR amplification of the pBS control plasmid containing or lacking HpaII site. Methylation levels were determined as the ratio of methylated or unmethylated rDNA relative to the respective input. Primer sequences used for ChIP-Chop assays are listed in Table EV2.

## Co-immunoprecipitation (co-IP)

Co-IP was performed as previously described, with minor modification (Cotney and Noonan, 2015; Li et al, 2024). Briefly, cells were crosslinked with 0.75% formaldehyde for 10 min, followed by quenching with 150 mM glycine for 10 min. Cross-linked cells were harvested by centrifugation at $2000 \times g$ for 5 min

at 4 °C and washed twice with cold PBS. Cells were lysed in ice-cold lysis buffer containing 10 mM Tris-HCl pH 8.0, 1 mM EDTA, 10% glycerol, 0.5% NP-40 and 0.25% Triton X-100, followed by sonication for 5 min (2 s on, 1 s off). Supernatant was collected by centrifugation at $20,000 \times g$ for 10 min at 4 °C, and protein concentration was determined using the BCA Protein Assay kit.

Primary antibodies were pre-incubated with protein A/G magnetic beads at room temperature for 2 h. Antibody-conjugated beads were then incubated with cell lysates overnight at 4 °C. After washing, the bound proteins were subjected to western blotting for interaction studies.

## Silver staining

Silver staining was performed as previously described (Zhang et al, 2021). Briefly, immunoprecipitates (IP) were subjected to SDS-PAGE, and the gels were fixed in a solution containing 50% ethanol, 12% acetic acid, and 0.05% formalin for 2 h, followed by three washes with 20% EtOH for 30 min each. The gels were then sensitized with 0.02% sodium thiosulfate for 2 min and rinsed with deionized water. Next, the gels were incubated in silver nitrate containing 0.076% formalin for 20 min, followed by two washes with deionized water. Development was carried out in sodium carbonate solution containing 0.0004% sodium thiosulfate and 0.05% formalin for 2 min, and the reaction was stopped using 12% acetic acids. The bound proteins were analyzed by mass spectrometry (MS, Applied Protein Technology, Shanghai, China). Proteins identified by MS are listed in Table EV3.

## RNA-seq and bioinformatic analysis

Total RNA was isolated from either mouse cortices or neurospheres cultured in vitro for 5 days using TRIzol, following the manufacturer's instructions. RNA samples were subjected to 150 nt pair end sequencing on the Illumina Novaseq platform by Annoroad Gene Technology (Beijing, China). Clean reads were aligned to the mouse genome (Mus_musculus.GRCm38.89.chr). Differentially expressed genes (DEGs) analysis were performed using DEGseq, with a cutoff of |Fold Change| ≥2 and FDR < 0.05. Gene Ontology (GO) analysis was carried out using DAVID (https://david.ncifcrf.gov/) or the Gene Ontology Consortium platform (http://geneontology.org/), and GO terms with $p < 0.05$ were considered significant. Heatmaps were generated using an online bioinformatic tool Heatmapper (http://www.heatmapper.ca/) (Babicki et al, 2016).

## Northern blot

Northern blot was performed as previously described (Li et al, 2018). DNA probes with biotin modifications were synthesized by General Biosystems (China). Total RNA samples and the RiboRuler High Range RNA Ladder were separated on either 1% formaldehyde agarose gel or 10% Urea-PAGE gels, then transferred and UV-crosslinked onto positively charged Hybond-N+ membranes. Hybridization with the biotinylated probes was carried out at 42 °C overnight. Signal detection was performed using HRP-conjugated streptavidin and visualized via enhanced chemiluminescence (ChemiScope, CLiNX). Immunoreactive bands were quantified by densitometric analysis using ImageJ software. Probe sequences are listed in Table EV2.

## Polysome profiling

Polysome profiling was carried out as previously described (Han et al, 2022), with minor modifications. 293T cells were treated with cycloheximide (Sigma) for 30 min at 37 °C, then harvested in lysis buffer containing 50 mM HEPES, 2 mM $MgCl_2$, 100 mM KCl, 1% Triton X-100, 10% Glycerol, proteinase inhibitor, 40 U/mL RNase inhibitor, and 1 mM DTT. Nuclei and mitochondria were removed by centrifugation at 14,000 rpm for 10 min at 4 °C. Ribosomal components, including 40S small subunit, 60S large subunit, 80S monosome, and polysome were separated using a 20–50% sucrose gradient via ultracentrifugation at 38,000 rpm for 3 h at 4 °C. Ribosome distribution was analyzed by measuring the absorbance of each fraction at 254 nm (A254) using UV photometry. Sucrose gradients were prepared using a Gradient Master™ (Biocomp), and fractionation was performed with a Piston Gradient Fractionator™ (Biocomp).

## Quantification and statistical analysis

All quantitative data represent the average of at least three biological replicates. Error bars indicate the standard error of the mean (SEM). Statistical significance was determined by Student's $t$ test or ANOVA in GraphPad Prism v10.1.2. $P < 0.05$ was considered statistically significant (indicated by an asterisk in the figures), $P < 0.01$ (indicated by two asterisks in the figures), $P < 0.001$ (indicated by three asterisks in the figures), and $P < 0.0001$ (indicated by four asterisks in the figures), while $P > 0.05$ was considered not significant (ns).

# Data availability

The raw sequencing data generated in this study have been deposited into the NCBI Sequence Read Archive (SRA) under the BioProject accession number PRJNA1073761 and BioProject accession number PRJNA1073873. The source data for this study are available in the following database record: BioStudies, accession number S-BIAD2101.

The source data of this paper are collected in the following database record: biostudies:S-SCDT-10_1038-S44319-025-00524-3.

# Peer review information

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

## Acknowledgements

This research was supported by the National Key R&D Program of China (2021YFA0804900 and 2020YFA0509300), the National Natural Science Foundation of China (82125009, 82330045, 82071185, 92149303, and 32121002), the Strategic Priority Research Program of the Chinese Academy of Sciences (XDB39000000), CAS Project for Young Scientists in Basic Research (YSBR-013), Plans for Major Provincial Science & Technology Projects (202303a07020004), Research Funds of Center for Advanced Interdisciplinary Science and Biomedicine of IHM (QYZD20220003), the Major Frontier Research Project of the University of Science and Technology of China (LS9100000002), Hefei Comprehensive National Science Center Hefei Brain Project, USTC Research Funds of the Double First-Class Initiative.

## Author contributions

**Wenbo Li**: Data curation; Formal analysis; Investigation; Methodology; Writing—original draft. **Juan Zhang**: Conceptualization; Formal analysis; Supervision; Funding acquisition; Writing—original draft; Writing—review and editing. **Qiang Liu**: Conceptualization; Resources; Supervision; Funding acquisition; Project administration; Writing—review and editing.

Source data underlying figure panels in this paper may have individual authorship assigned. Where available, figure panel/source data authorship is listed in the following database record: biostudies:S-SCDT-10_1038-S44319-025-00524-3.

## Disclosure and competing interests statement

The authors declare no competing interests.

# Expanded View Figures

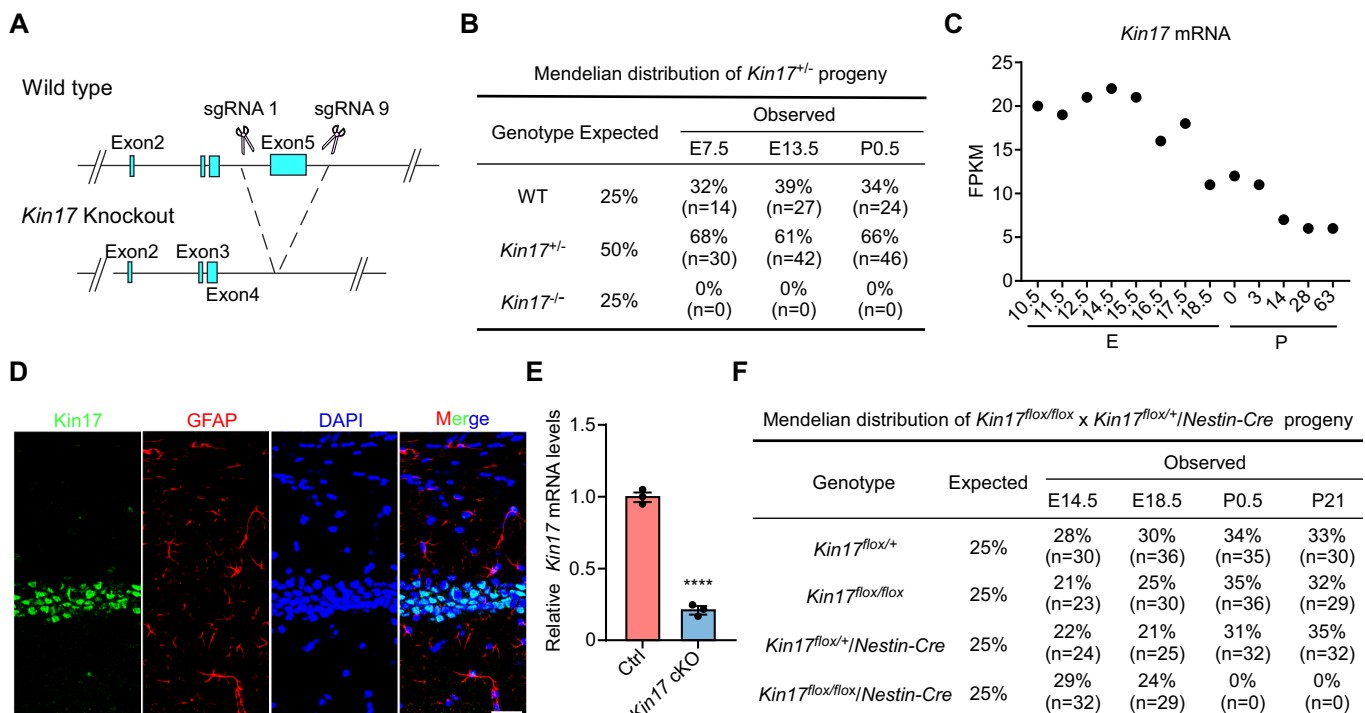

**Figure EV1. Mice with Kin17 depletion in neuronal progenitor cells show embryonic survival.**

(A) Generation of *Kin17* knockout mice. Schematic of the genomic target sites in *Kin17* locus. (B) Genotype distribution of embryos produced from mating of *Kin17*[+/−] mice at E7.5, E13.5, and P0.5. (C) Scatter plot showing Kin17 expression in mouse brains at embryonic and postnatal stages, based on data from the EMBL-EBI Expression Atlas. (D) Representative immunofluorescence images of Kin17 (green), GFAP (red), and DAPI (blue) in 3-month-old WT brain. Scale bar, 30 μm. (E) Kin17 mRNA levels in the cortex of control and *Kin17* cKO mice at E14.5, by qPCR ($P < 0.0001$). Data were normalized against β-actin ($n = 3$). (F) Genotype distribution of embryos and offspring from mating of *Kin17*[flox/flox] and *Kin17*[flox/+/Nestin-Cre] mice, assessed at E14.5, E18.5, P0.5, and P21. Statistical analysis was performed using two-tailed *t* test. Error bars denote the SEM. ****$P < 0.0001$.

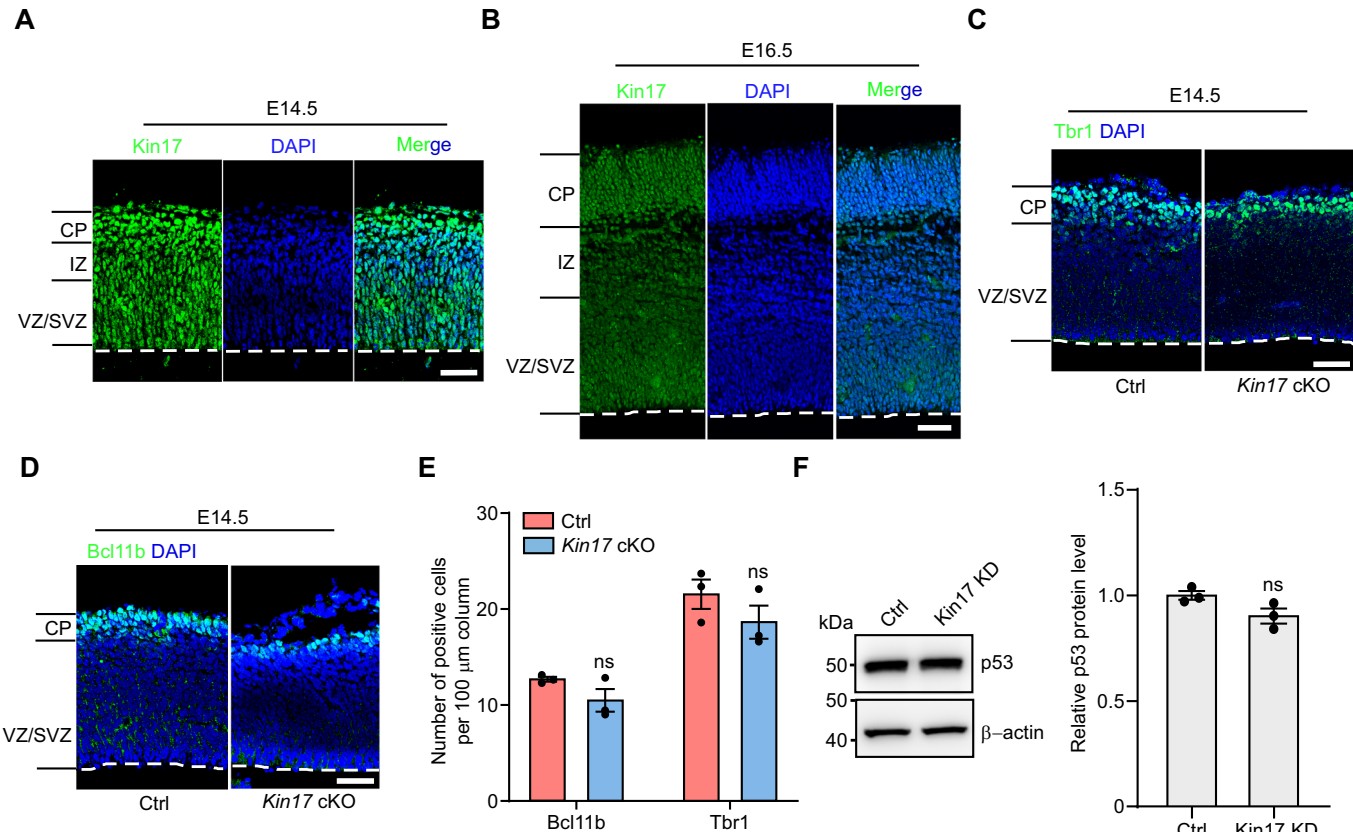

**Figure EV2.  Kin17 depletion in NPCs leads to p53-independent apoptosis.**

(A, B) Representative immunofluorescence images of Kin17 (green) and DAPI (blue) in the cerebral cortex of WT mice at E14.5 (A) and E16.5 (B). Scale bar, 50 μm. (C, D) Representative immunofluorescence images of Tbr1 (green) and DAPI (blue) (C), of Bcl11b (green) and DAPI (blue) (D), in control and Kin17 cKO brain at E14.5. Scale bar, 50 μm. (E) Quantification of Tbr1 ($P = 0.1612$) or Bcl11b ($P = 0.2613$) positive cells in both control and Kin17 cKO brains ($n = 3$ per group). (F) p53 protein level in control and Kin17 KD 293T cells, by immunoblotting and densitometric analysis ($P = 0.0763$) ($n = 3$). β-actin was included as a loading control. Statistical analysis was performed using two-tailed $t$ test (E, F). Error bars denote the SEM. ns not significant.

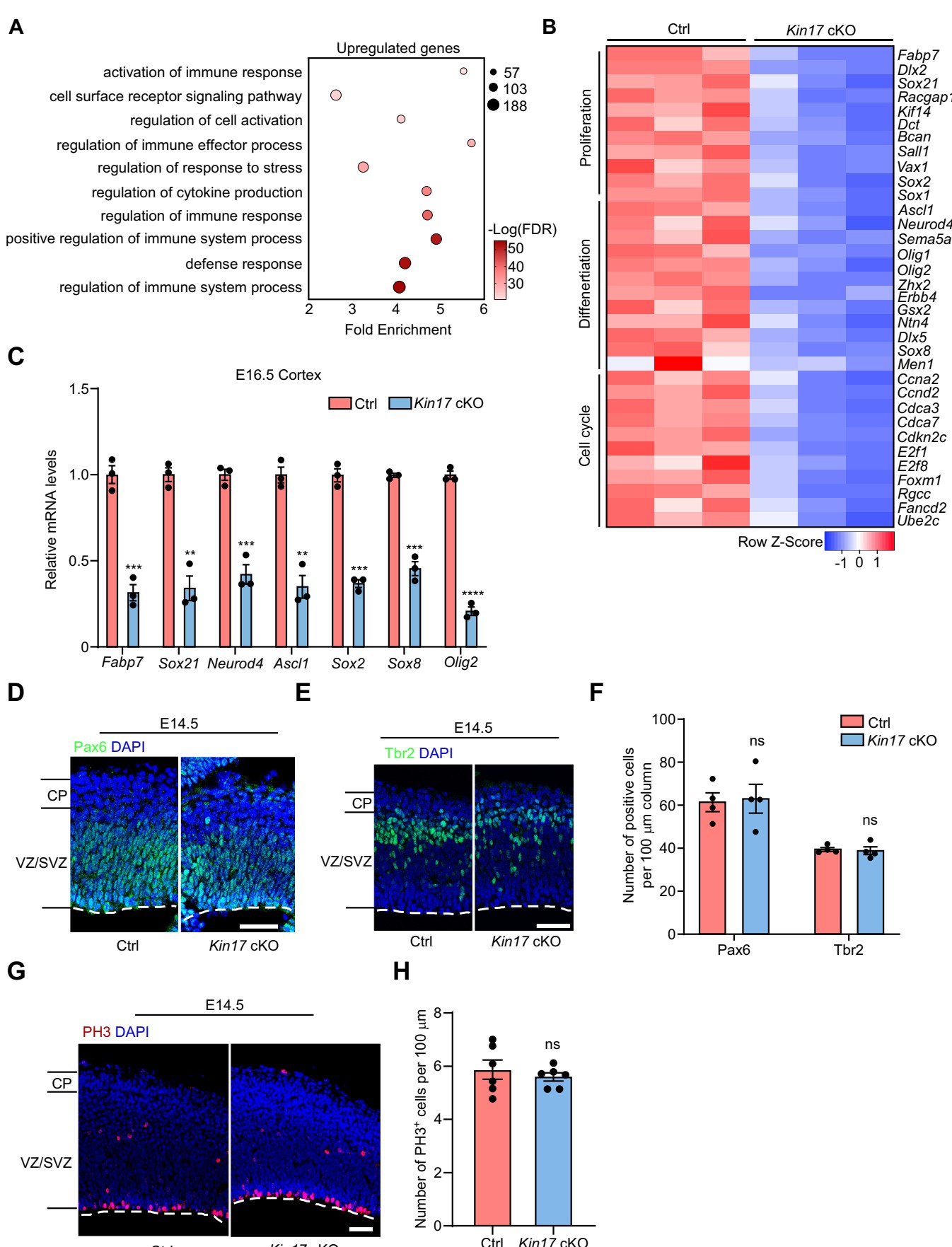

**Figure EV3. Defective cortical lamination becomes evident at E16.5.**

(A) A bubble plot showing GO analysis of upregulated genes in the cerebral cortex of *Kin17* cKO versus control mice at E16.5. (B) A heatmap showing expressions of genes related to "proliferation", "differentiation", and "cell cycle" in the cerebral cortex of *Kin17* cKO versus control mice at E16.5 ($n = 3$ per group). (C) Levels of genes related to "proliferation" and "differentiation", by qPCR (Fabp7, $P = 0.0006$; Sox21, $P = 0.0013$; Neurod4, $P < 0.0009$; Ascl1, $P = 0.0013$; Sox2, $P = 0.0001$; Sox8, $P = 0.0002$; Olig2, $P < 0.0001$). Data were normalized against β-actin ($n = 3$). (D) Representative immunofluorescence images of Pax6 (green) and DAPI (blue) in the cerebral cortex of control and *Kin17* cKO mice at E16.5. Scale bar, 50 μm. (E) Representative immunofluorescence images of Tbr2 (green) and DAPI (blue) in the cerebral cortex of control and *Kin17* cKO mice at E16.5. Scale bar, 50 μm. (F) Quantitative analysis of Pax6 ($P = 0.778$) or Tbr2 ($P = 0.8618$) positive cells in control and Kin17 cKO cortices at E14.5 ($n = 3$ per group). (G) Representative immunofluorescence images of PH3 (red) and DAPI (blue) in the cerebral cortex of control and *Kin17* cKO mice at E14.5. Scale bars, 50 μm. (H) Quantitative analysis of PH3-positive cells in control and *Kin17* cKO mice ($P = 0.539$) ($n = 6$ per group). Statistical analysis was performed using two-tailed *t*-test. Error bars denote the SEM (C, F, H). **$P < 0.01$; ***$P < 0.001$; ****$P < 0.0001$; ns not significant.

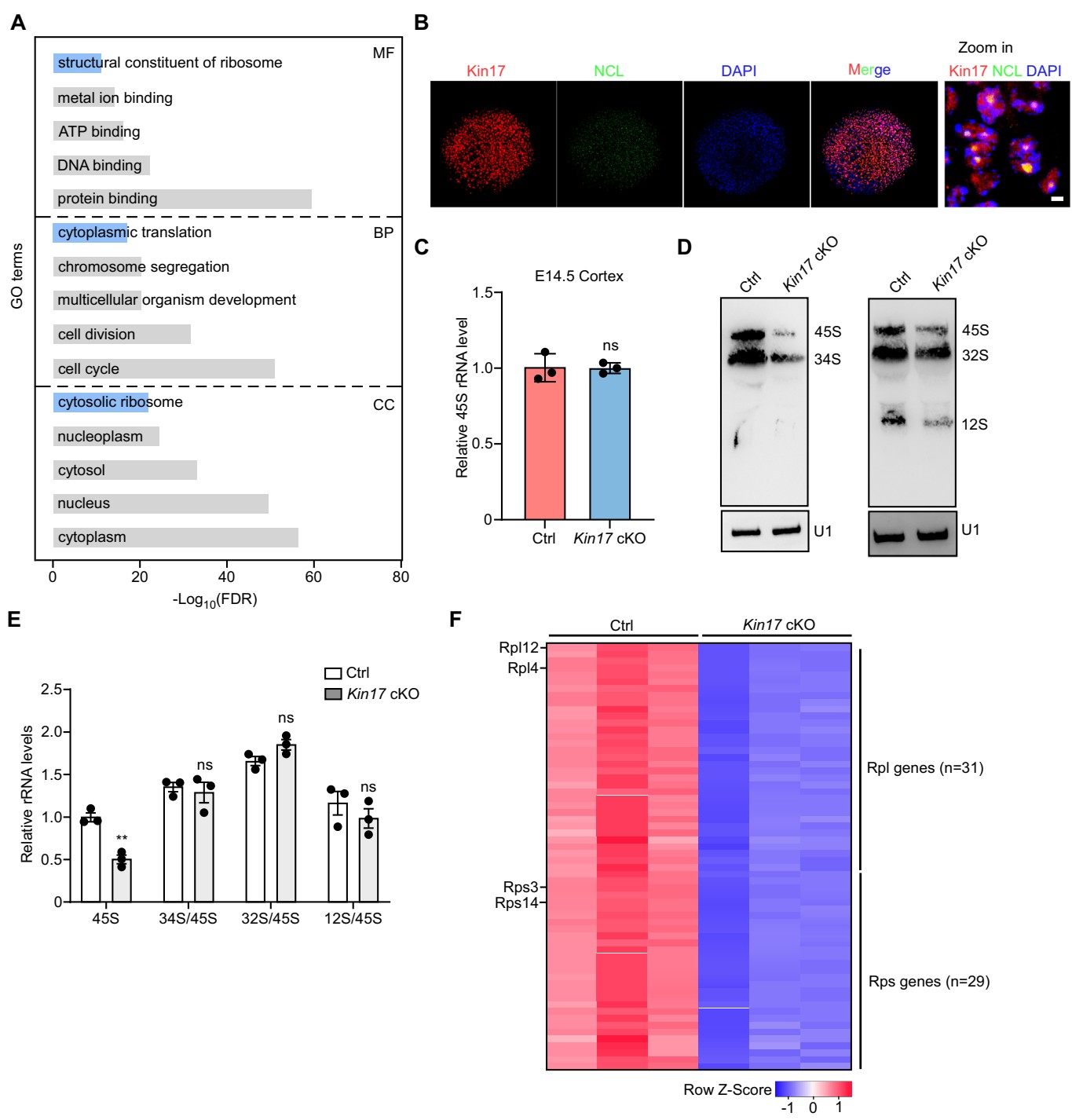

**Figure EV4. Kin17 has no impact on rRNA processing.**

(A) GO analysis of shared genes by molecular function (MF), biological process (BP) and cellular component (CC), highlighting enriched biological processes. (B) Representative immunofluorescence images of Kin17 (red), NCL (green), and DAPI (blue) in cultured neurospheres. Scale bar, 100 μm (left), 3 μm (right). (C) 45S rRNA level was determined in control and Kin17 cKO cortices at E14.5, by qPCR analysis ($P = 0.9689$) ($n = 3$). (D, E) Levels of 45S pre-rRNA, intermediate processed product 34S, 32S, and 12S in control and Kin17 cKO cortices were analyzed by northern blot (D) and densiometric analysis (E) ($n = 3$) (45S, $P = 0.0026$; 34S/45S, $P = 0.647$; 32S/45S, $P = 0.0722$; 12S/45S, $P = 0.3844$). U1 was included as a loading control. (F) A heatmap demonstrating differentially expressed ribosome protein genes in Kin17 cKO versus control neurospheres. Statistical analysis was performed using two-tailed Student's t test (C, E). Error bars denote the SEM. ns no significant.

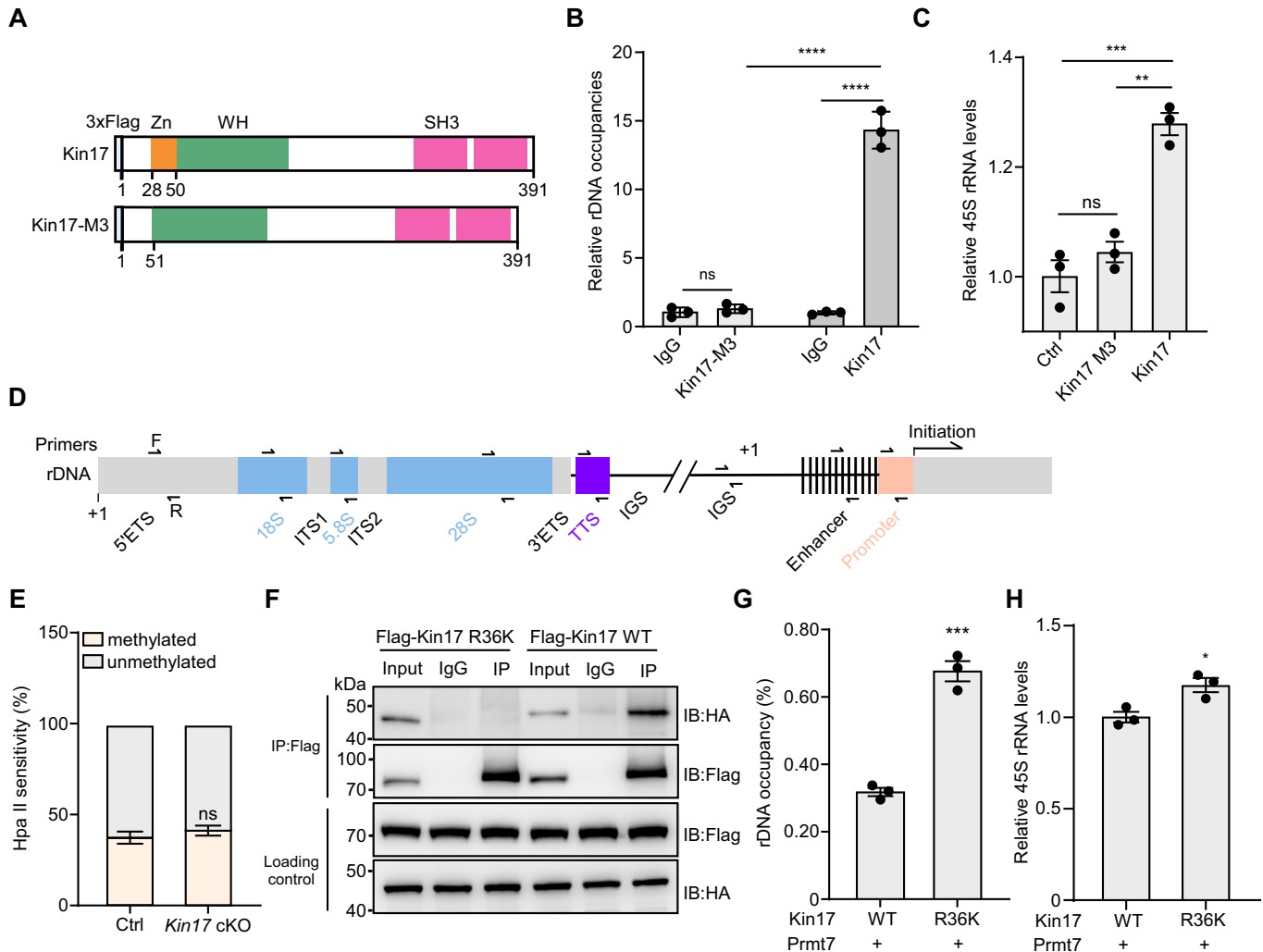

**Figure EV5. DNA binding domain is required for Kin17's binding to rDNA.**

(A) Schematic showing the generation of wild-type Kin17 (Kin17-WT) and the zinc finger domain-deleted mutant (Kin17-M3), each tagged with three Flag epitopes on the N-terminus. (B, C) N2a cells were transfected with either Kin17-M3 or Kin17-WT plasmids for expression. (B) Relative occupancy at the rDNA promoter was assessed by ChIP-qPCR using an anti-Flag antibody, followed by qPCR detection ($n = 3$) (Kin17-M3 versus IgG, $P = 0.3844$; Kin17 versus IgG, $P < 0.0001$; Kin17 versus Kin17-M3, $P < 0.0001$) (C) 45S pre-rRNA level was determined by qPCR-based analysis ($n = 3$). (Kin17-M3 versus control, $P = 0.6814$; Kin17 versus control, $P = 0.0005$; Kin17 versus Kin17-M3, $P = 0.0012$) (D) Schematic illustrating the primer design strategy for detecting the enhancer, promoter, 5′ external transcribed spacer (5′ETS), 18S, 28S, 5.8S, internal transcribed spacer (ITS), intergenic spacer (IGS), and transcription termination site (TTS) of rDNA. (E) Distribution of methylated and unmethylated rDNA in control or Kin17 cKO neurospheres ($n = 3$). (F–H) N2a cells were transfected with either HA-tagged WT Kin17 or R36K mutant, along with Flag tagged Prmt7 for expression. (F) The interaction of Kin17 and PRMT7 was assessed by co-IP using an anti-Flag antibody, followed by immunoblotting with an anti-HA antibody. (G) rDNA occupancy of Kin17 was determined by ChIP-qPCR analyses using an anti-HA antibody ($n = 3$) ($P = 0.0002$). (H) 45S pre-rRNA levels were determined by qPCR-based analyses ($n = 3$) ($P = 0.02$). Statistical analysis was performed using one-way ANOVA (C) with Bonferroni correction or two-tailed Student's t test (B, E, G, H). Error bars denote the SEM. *$P < 0.05$, **$P < 0.01$, ***$P < 0.001$; ****$P < 0.0001$; ns not significant.

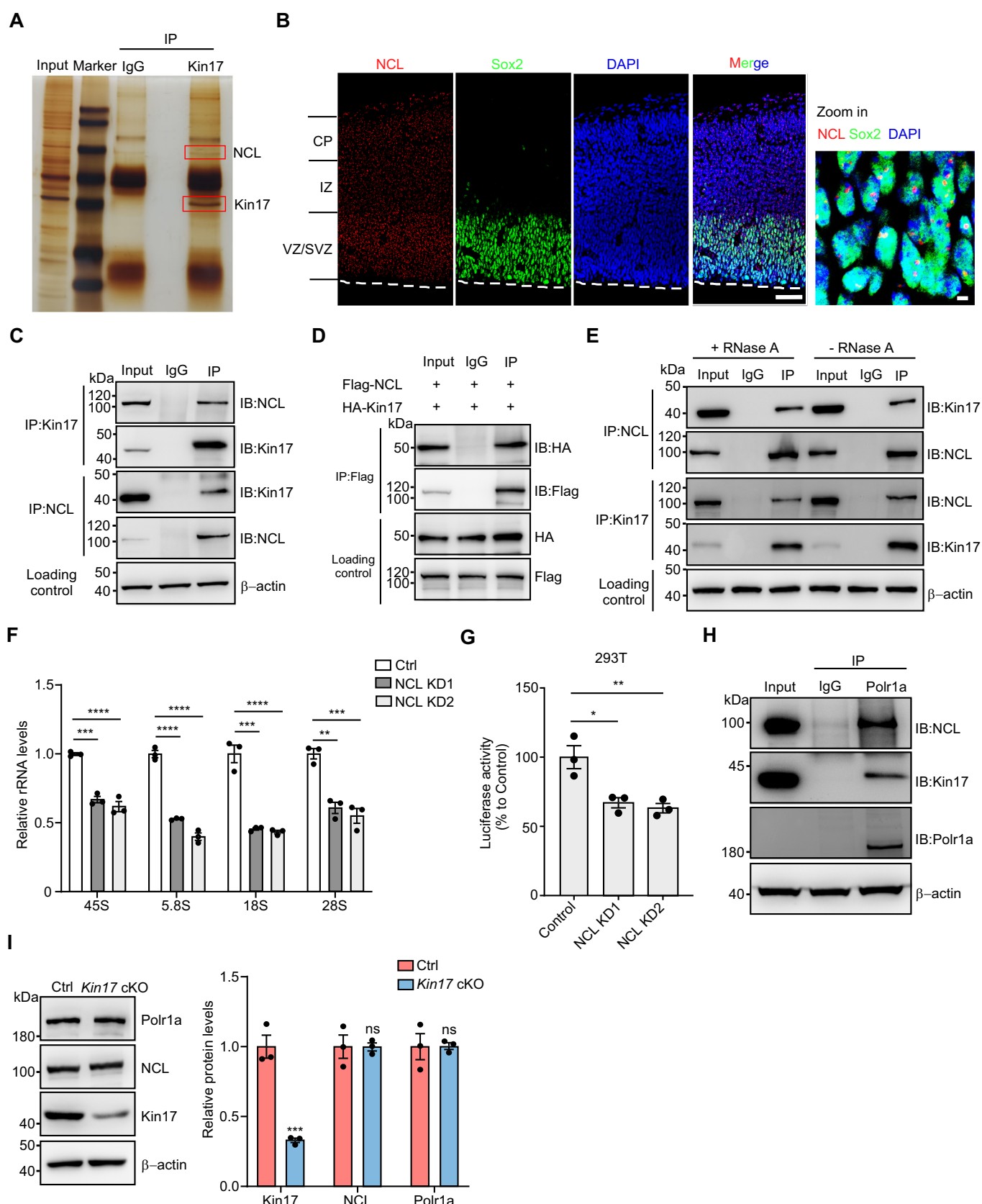

◀ **Figure EV6.  Kin17-NCL interaction is required to promote rDNA transcription.**

(A) Kin17-associated proteins were separated by electrophoresis and visualized by silver staining. (B) Representative immunofluorescence images of NCL (red), Sox2 (green), and DAPI (blue) in the cerebral cortex of WT mice at E16.5. Scale bar, 50 μm (left), 3 μm (right). (C) The interaction of endogenous Kin17 and NCL in the cortex of WT mice at E16.5, assessed by co-IP using an anti-Kin17 antibody followed by immunoblotting detection with an anti-NCL antibody, and reciprocally by co-IP with an anti-NCL antibody followed by immunoblotting detection with an anti-Kin17 antibody. (D). The interaction of exogenous HA-Kin17 and Flag-NCL in 293 T cells transfected with HA-Kin17 and Flag-NCL for expression, determined by co-IP with an anti-Flag antibody and immunoblotting detection with an anti-HA antibody ($n = 3$). (E) Neurosphere lysates were treated with either RNase A or a control solvent, followed by co-IP using an anti-NCL antibody. The NCL-associated Kin17 was detected by immunoblotting with an anti-Kin17 antibody. Reciprocally, co-IP was performed using an anti-Kin17 antibody, and the Kin17-associated NCL was similarly detected by immunoblotting with an anti-NCL antibody. (F) The levels of rRNA (45S, 5.8S, 18S, and 28S) in NCL KD neurospheres, by qPCR ($n = 3$ per group) (45S: NCL KD1, $P = 0.0002$; NCL KD2, $P < 0.0001$; 5.8S: NCL KD1, $P < 0.0001$; NCL KD2, $P < 0.0001$; 18S: NCL KD1, $P = 0.0001$; NCL KD2, $P < 0.0001$; 28S: NCL KD1, $P < 0.0017$; NCL KD2, $P < 0.0008$). Data were normalized against U1 snRNA. (G) rDNA promoter reporter plasmid was transfected into control or NCL KD 293 T cells. Luciferase activity was determined by Dual-Luciferase reporter assay ($n = 3$ per group). (H) The interaction of Polr1a with NCL or Kin17 in neurospheres, determined by co-IP with an anti-Polr1a antibody, followed by immunoblotting detection with an anti-NCL antibody or an anti-Kin17 antibody ($n = 3$ per group). (I) Polr1a and NCL expression in control and Kin17 cKO neurospheres, by immunoblotting and densitometry ($n = 3$ per group) (Kin17, $P = 0.0013$; NCL, $P = 0.9818$; Polr1a, $P = 0.9477$). Statistical analysis was performed using one-way ANOVA with Bonferroni correction (F, G) or two-tailed Student's $t$ test (I). Error bars denote the SEM. *$P < 0.05$; **$P < 0.01$; ***$P < 0.001$; ****$P < 0.0001$; ns not significant.

