## [Peer Review File · EMBO Reports]

Kin17 promotes rDNA transcription, ribosomal biogenesis, and cortical lamination

Qiang Liu, Juan Zhang, and Wenbo Li

Corresponding author(s): Qiang Liu (liuq2012@ustc.edu.cn) , Juan Zhang (zj2014@ustc.edu.cn)

Review Timeline:

Submission Date:	26th Jun 24
Editorial Decision:	15th Aug 24
Appeal Received:	25th Apr 25
Editorial Decision:	2nd Jun 25
Revision Received:	24th Jun 25
Accepted:	30th Jun 25

Editor: Esther Schnapp / Martina Rembold

Transaction Report:

Dear Dr. Liu

Thank you for the submission of your research manuscript to EMBO Reports and for your patience while it was under review. We have now received the three enclosed reports on it. Since my colleague Esther Schnapp is currently traveling, I have stepped in as the secondary editor for your manuscript.

I am sorry to say, that the evaluation of your manuscript is not a positive one. As you will see, although the referees acknowledge that the findings are potentially interesting, they raise important, partially overlapping criticisms of the manuscript and consider the data insufficient to support the main conclusions. The referees raise concerns regarding the pleiotropic effect of Kin17 LOF and whether the observed phenotypes can be attributed to its proposed effects on rDNA transcription and ribosomal biogenesis.

Given these concerns, the amount of work required to address them, the uncertain outcome of these experiments, and the fact that EMBO reports can only invite revision of papers that receive enthusiastic support from a majority of referees, I am sorry to say that we cannot offer to publish your manuscript.

I am sorry to disappoint you on this occasion, and hope that the referee comments will be helpful in your continued work in this area.

Yours sincerely

Referee #1:

In this manuscript Li et al. investigate the role of the DNA binding protein Kin17 during brain development, and identify a novel function of this protein in the regulation of cortical lamination, as well as in the regulation of rDNA transcription and protein synthesis. To this end, the Authors analyze conditional knock-out (Kin17 cKO) mice during development and neurospheres derived from the same mice.

The study is very well written and relevant, and the findings are novel, accurately reported and based on a robust set of data. In my view the most interesting aspect is the evidence of the role of Kin17 on rRNA synthesis along with the impact on cerebral cortical lamination. However, to better understand these observations, I would strongly recommend to include the following analysis and discussion before acceptance for publication.

1) The Authors write "We observed that the absence of Kin17 reduces the neuronal numbers in both deep-layer (layer V and layer VI), upper-layer II/III, as well as layer IV, impairing the laminated structure of cerebral cortex." It is unclear whether the expression of Kin7 protein in the mouse cortex during embryonic development is more typical of particular layers / regions in the developing cortex. The Authors should provide immunofluorescence analysis of Kin17 protein expression in mouse cortical brain sections at different stages, to understand whether there might be a layer-specific expression.

2) The Kin17cKo shows a decreased activity of RNA Pol I and reduced expression of some ribosomal proteins. The Authors write: "Furthermore, we measured the levels of ribosomal proteins, including large subunit protein Rpl12 and Rpl4, and small subunit protein Rps3 and Rps14. We observed a significant decrease in the levels of these ribosomal proteins in Kin17 cKO neurospheres compared to control neurospheres (Fig 4F), suggesting impaired ribosome biogenesis in the absence of Kin17." The rationale for analyzing the indicated proteins is not specified. Next, it would be important to understand whether the effect of cKO is limited to these proteins or it is more general. It would be also important to explain why the synthesis of these (and others) ribosomal proteins is decreased, because it seems that Kin17 regulates rRNA synthesis. Moreover, I am not convinced that these experiments show impaired ribosome biogenesis, it is still possible that ribosomes are formed but maybe they are defective. This point should be better explained.

3) The Authors also write "we demonstrate that Kin17 promotes rDNA transcription through recruiting NCL to increase the loading of Pol I complex on the rDNA promoter, consequently promotes ribosomal biogenesis." If the phenotype is to be ascribed to the impact of Kin17 with NCL and/or PolI, would the loss of NCL and / or PolI activity per se lead to similar phenotypes as the Kin17 cKO in cortical development? What would be the phenotype of cKO mice lacking factors important for RNA Pol I and rRNA synthesis in neural progenitors? The analysis and/or discussion of these questions would help to better understand the specificity of the function of Kin17 on cerebral cortical lamination.

4) I also recommend the Authors to include the analysis of apoptosis and/or cell death to gain more insights about the fate of the defective neurons in the developing brain.

5) Finally, it would be important to gain some insights about the impact of loss of Kin17 on the nuclear localization of nucleolin as well as other nucleolar proteins in the Kin17 cKO mouse cortex, to understand whether this results in nucleolar stress, and

(also linked to point 4) in increased p53 levels, and apoptosis.

6) The Authors should also provide some evidence of interaction of Kin17 with NCL not only in the neurospheres, but also in tissue sections.

Referee #2:

Comments for the Authors:

The study entitled "Kin17 promotes cortical lamination through eliciting rDNA transcription and ribosomal biogenesis in mice" by Li et al. examines the role of Kin17 in cortical lamination. The authors indicate that Kin17 cKO mice display a notable reduction in cortex size and cell numbers. They attempt to link these phenotypic changes to ribosome biogenesis through recruiting NCL and Pol I to the rDNA promoter. It is essential to determine the distinct contributions of NCL and Polr1a in facilitating rDNA transcription and how their interactions with Kin17 enhance ribosomal biogenesis in NPCs. Some experiments were well conducted, but several issues need to be addressed. There are many experiments and data poorly explained, the order of the figures lack logic and most significantly the conclusions are not justified by the data. In the following part my major and minor concerns are listed in more detail:

1. My primary concern is that the depletion of a crucial factor like Kin17 can lead to many functional consequences in general, it is not solely specific to NPCs proliferation and differentiation. It is not clear what cells in the brain express Kin17 and the factor impacts various other cellular processes and functions. What other nerve subtype cells express it? Why then do you think that Kin17 is affecting neurons only? In your paper's figures, there are also no clear immunofluorescence staining images demonstrating the expression of Kin17 in neural stem cells.
2. On the one hand depletion of Kin17 in NPCs results in brain atrophy, including NPC proliferation (the number of PH3-positive cells and BrdU-positive cells) and NPC differentiation (*Tbr1*, *Bcl11b*, *Satb2*) at E16.5 stage, but on the other hand, these changes in the proliferation and differentiation of neural stem cells are not observed at the E14 stage. Considering that neural development is a sequential process, and the expression level of Kin17 protein is significantly higher at the embryonic E14 stage compared to the E16 stage (Fig 1A). It is perplexing why the knockout of Kin17 protein affects brain development only at the E16 stage. Furthermore, your experimental data do not support why this phenomenon occurs. What is the molecular mechanism underlying this discrepancy?
3. Why did the author choose neurospheres as the material for RNA sequencing? The in vitro culture conditions differ significantly from the in vivo physiological state. Using in vivo tissue materials would better reflect the causes of these differences. It is recommended that the authors perform joint analysis with in vivo tissue sequencing.
4. The authors claim that Kin17 binds to rDNA, and then regulate its transcriptional activity, but proper controls for such a statement are missing. Moreover, it is not easy to understand why the authors focused on NCL in their further analyses. The authors should provide more compelling evidence that NCL is actually expressed in NPCs.

Minor comments

1. The quality of some figure images (such as Fig 1J) needs to be improved. Furthermore, the patterns of *Tbr1* (a specific marker for layer VI) immunofluorescence staining at E16.5 stage differ from that reported in previous studies (for instance Mihalas et al., 2016, Cell Reports 16, 92-105 June 28).
2. Page 6, "Immunostaining conducted 48 hours post-BrdU injection showed a notable decrease in the number of cells positive for BrdU but negative for Ki67 (BrdU+/Ki67-) in Kin17 cKO mice at E15.5 compared to control mice (Figs 2J and 2K)". The immunofluorescence staining images are not very clear in Fig 2J. There should be double-positive cells (BrdU+/Ki67+) in addition to the negative cells (BrdU+/Ki67-).
3. It is not clear, why the authors look at NCL, when they correlate Kin17 with rDNA promoter sequence. There is a lack of direct experimental evidence to support the rationale behind this choice.
4. The manuscript would benefit from careful reading for grammatical errors. At some stages the manuscript is difficult to read, or does not make sense.

Overall, this work needs a lot of extra analysis in order to provide new novel findings that go significantly beyond addressing the issues I raised above.

Referee #3:

The authors study the role of Kin17 in NPC, suggesting that the protein plays an important role in development. They suggest that Kin17 activates rDNA transcription through interaction with Nucleolin, thereby promoting ribosome biogenesis.

In my opinion, the results described require a more detailed functional link, showing the involvement of Kin17 in rDNA regulation. Due to its many functions in replication, repair, splicing, mRNA processing, the effects described in this manuscript could also be

indirect. Especially the work with the highly repetitive rDNA genes requires additional controls, as these genes are difficult to analyze in ChIP experiments. Additionally, rRNA transcription is modulated by any kind of changes in the cellular status and by indirect effects that are expected by a factor altering the expression of many genes.

Furthermore, before starting my review of the experimental data, I have to say that the authors willingly left out the known fact that Kin17 plays an important role in ribosome biogenesis. The authors cite the paper 34449532, but only mention the spliceosome interaction. The manuscript 34449532 shows a BioID, revealing the interactors of Kin17, clearly showing the direct interaction with the SSU processome and being involved in 60S maturation. Furthermore, no interaction with Nucleolin was detected in this study. In the light of this knowledge, the novelty has to be re-evaluated, and the controls to distinguish rRNA transcription and processing effects have to be much better controlled than presented here. rRNA processing and transcription of rDNA genes are intimately linked and influence each other.

If Kin17 regulates the rRNA genes, the protein should be located in the nucleolus. It would be good to show imaging data with the localization of Kin17 with appropriate nucleolar markers.

If Kin17 functions as an activator of rDNA transcription, it should bind to the active rDNA genes that are not methylated. An experiment, called ChIP-Chop (24162999) can distinguish between binding of the protein to the active (non-methylated) and inactive (methylated) rDNA genes. This would first, show a specific targeting to rDNA and provide a clear link to its suggested activating function.

A ChIP experiment combined with qPCR, testing only one locus in rDNA, having no positive controls (UBF or TTF) is not sufficient. At least IGS regions of rDNA, enhancer and coding sequences have to be tested with appropriate controls. As mentioned above a ChIP-Chop assay to reveal the co-localization of Kin17 with active genes is also required. By the way, the exact primer sequences, positions within rDNA, DNA fragment length, linearity of qPCR are information and parameters that are not provided in the text and supplemental information. This is required.

Transfecting the zinc-finger domain mutation of Kin17 that should not bind to DNA anymore would also be a good control to evaluate the binding of Kin17 to rDNA and its transcriptional effects. Furthermore, it was shown that Kin17 interacts with Prmt7 and that protein methylation releases it from chromatin. Including a functional mutant of Kin17 would be important to separate individual biological processes.

The authors suggest the interaction of Kin17 with Nucleolin through its RNA binding domain. Since the nucleolus is a compartment with high RNA density, and Nucleolin is also an RNA binding protein, it would be important to know whether the interaction is direct or indirectly mediated through RNA. To reveal the specificity of this interaction I recommend to either use purified recombinant proteins to repeat the interaction study, or to include RNase treatment in the IP conditions.

The RNA-seq data affects many genes. The authors should survey the effects on known Polr regulators. For example TIF1A, has a similar effect and is embryonic lethal. A downregulation of this factor would have a similar effect.

What about the known effects on the replication machinery. The effects would be the same. No proliferation of the cells, as observed. How do the authors rule out this kind of effects. The same with the Ribosome processing effects of Kin17 that were missed to mention. How can the authors be sure that the observed effects are not due to a shut down of rRNA transcription by defects in rRNA processing?

I think there are many open questions and experiments to be performed to prove a direct link between Kin17 and rDNA gene activation. In this form I do not recommend publication.

** As a service to authors, EMBO Press provides authors with the ability to transfer a manuscript that one journal cannot offer to publish to another journal, without the author having to upload the manuscript data again. To transfer your manuscript to another EMBO Press journal using this service, please click on Link Not Available

April 25, 2025

Martina Rembold, Ph.D.
Senior Editor
EMBO Reports

Dear Dr. Rembold,

Thank you for the review of our manuscript entitled “Kin17 promotes cortical lamination through eliciting rDNA transcription and ribosomal biogenesis in mice” (EMBOR-2024-59864V1).

We have made substantial revisions to the manuscript in response to the Reviewers' concerns and suggestions. These include the addition of numerous new experiments, expanded discussions, and clarifications throughout the text. We believe these changes have substantially strengthened the manuscript and would like to have this revised manuscript reconsidered for publication in *EMBO Reports*.

Please find attached for the revised version of the manuscript along with a detailed point-by-point response letter to all Reviewer comments. We have carefully addressed each concern and believe that the revised version satisfactorily meets the expectations for publication in *EMBO Reports*.

Thank you again for your time and consideration. We look forward to your response.

Respectfully,

Qiang Liu, Ph.D
Professor, School of Life Science
University of Science & Technology of China

Reviewer #1:

In this manuscript Li et al. investigate the role of the DNA binding protein Kin17 during brain development, and identify a novel function of this protein in the regulation of cortical lamination, as well as in the regulation of rDNA transcription and protein synthesis. To this end, the Authors analyze conditional knock-out (Kin17 cKO) mice during development and neurospheres derived from the same mice.

The study is very well written and relevant, and the findings are novel, accurately reported and based on a robust set of data. In my view the most interesting aspect is the evidence of the role of Kin17 on rRNA synthesis along with the impact on cerebral cortical lamination. However, to better understand these observations, I would strongly recommend to include the following analysis and discussion before acceptance for publication.

We thank the Reviewer for the positive comments, as quoted here “The study is very well written and relevant, and the findings are novel, accurately reported and based on a robust set of data”. We provide our full point-by-point responses below.

1) The Authors write "We observed that the absence of Kin17 reduces the neuronal numbers in both deep-layer (layer V and layer VI), upper-layer II/III, as well as layer IV, impairing the laminated structure of cerebral cortex." It is unclear whether the expression of Kin17 protein in the mouse cortex during embryonic development is more typical of particular layers / regions in the developing cortex. The Authors should provide immunofluorescence analysis of Kin17 protein expression in mouse cortical brain sections at different stages, to understand whether there might be a layer-specific expression.

RESPONSE: We have now examined Kin17 expression in the developing cortex of WT mice at E14.5 and E16.5 using immunofluorescence analysis. Representative images show that Kin17 is ubiquitously expressed in all cortical layers at both different development stages (Response Document Figure 1A-B), indicating that Kin17's role in cortical development is not restricted to specific layers or regions. We have now included those newly generated data in the revised manuscript (page 5 line 135-137).

Response Document Figure 1

Response Document Figure 1. Representative immunofluorescence images of Kin17 (green) and DAPI (blue) in cerebral cortex of WT mice at E14.5 (A) and E16.5 (B). Scale bar, 50 μ m.

2) The Kin17 cKO shows a decreased activity of RNA Pol I and reduced expression of some ribosomal proteins. The Authors write: "Furthermore, we measured the levels of ribosomal proteins, including large subunit protein Rpl12 and Rpl4, and small subunit protein Rps3 and Rps14. We observed a significant decrease in the levels of these ribosomal proteins in Kin17 cKO neurospheres compared to control neurospheres (Fig 4F), suggesting impaired ribosome biogenesis in the absence of Kin17." The rationale for analyzing the indicated proteins is not specified. Next, it would be important to understand whether the effect of cKO is limited to these proteins or it is more general. It would be also important to explain why the synthesis of these (and others) ribosomal proteins is decreased, because it seems that Kin17 regulates rRNA synthesis. Moreover, I am not convinced that these experiments show impaired ribosome biogenesis, it is still possible that ribosomes are formed but maybe they are defective. This point should be better explained.

RESPONSE: We thank the Reviewer for focusing our attention on this issue. Among ribosomal proteins, Rpl12, Rpl4, Rps3, and Rps14 exhibited the most pronounced reductions in Kin17 cKO neurospheres compared to controls, as demonstrated in our RNA-seq data (Response Document Figure 2A). Based on this, we selected these representative ribosomal proteins for further validation and confirmed their downregulation in Kin17 cKO neurospheres.

Of note, our RNA-seq data revealed that a total of 31 Rpl genes and 29 Rps genes were significantly downregulated in Kin17 cKO neurospheres compared to controls (Response Document Figure 2A), suggesting that the effect of Kin17 depletion is not limited to a few ribosomal proteins but reflects a global reduction in ribosomal protein

expression. We have included these data in the revised manuscript (page 9 line 263-267).

Ribosome biogenesis requires precise coordination between the production of rRNA and ribosomal protein. Rrn3 (also known as TIF1-A in vertebrates) protein is known to bind RNA polymerase I (RNAPI) and activates rDNA transcription by facilitating its recruitment to the 35S rDNA gene promoters. In addition, Rrn3 simultaneously regulates rRNA and ribosomal protein gene (RPG) transcription following inhibition of target of rapamycin complex 1 (TORC1) kinase, when ribosome biogenesis is strongly downregulated. This trans-effect on RNPII is specific to RPGs, providing strong evidence for a regulatory crosstalk between RNAPI and RNAPII that coordinates rRNA and RPG transcription (PMID:16882981).

RPG transcription is also governed by transcriptional activator Ifh1, which cooperates with its promoter-bound partner Fhl1. Under optimal growth condition, Ifh1 binds to multiple RPG promoters to enhance their expression. Upon TORC1 inhibition or nutrient deprivation, Ifh1 is rapidly released from RP promoters. This release is dependent on Utp22, which sequesters Ifh1 activity and prevents its binding to RPG promoters. Importantly, Utp22-mediated inhibition of Ifh1 activity is inversely proportional to RNAPI activity, supporting a mechanistic link between rRNA production and RPG transcription (PMID:27818142). Additionally, Ifh1 promoter binding is coordinated with RNA polymerase I activity upon prolonged TORC1 inhibition to maintain a balance of RP and rRNA production (PMID:15616569). Moreover, Ifh1 is a component of CURI complex, which includes several proteins involved in pre-rRNA processing, suggesting a role of Ifh1 in coupling RPG transcription and pre-rRNA processing (PMID:17452446). Taken together, these lines of evidence demonstrate that rRNA production and RPG transcription are tightly linked to maintaining a balanced ribosome biogenesis.

We have expanded our discussion regarding this issue in our revised manuscript (page 14 line 447-469).

To assess the impact of Kin17 on ribosome formation, we have now conducted polysome profiling to separate and analyze the small subunits (40S), large subunits (60S), monosomes (80S), and polysomes (mRNA-ribosome complexes with multiple ribosomes). We observed a significant reduction in 40S, 60S, 80S monosomal, and polysomes in Kin17 KD 293T cells compared to control 293T cells (Response Document Figure 2B-C). Of note, the ratios of 40S and 60S to 80S did not differ

between control and Kin17 KD 293T cells, indicating that Kin17 has no impact on monosome assembly (Response Document Figure 2D). Similarly, the monosome-to-polysome ratio was comparable between control and Kin17 KD cells (Response Document Figure 2D), suggesting that polysomal formation is also unaffected. These newly generated data support a role of Kin17 in ribosome biogenesis, specifically influencing the overall production of ribosomal subunits without impairing their assembly into functional ribosomes. We have included these data in the revised manuscript (page 9 line 270-280).

Response Document Figure 2

Response Document Figure 2. (A) A heatmap demonstrating differentially expressed ribosome protein genes in Kin17 cKO versus control neurospheres. (B) Cytoplasmic ribosome distribution in control (red) and Kin17 KD (blue) 293T cells (n=3). (C) Quantification of area under curve (AUC) for 40S, 60S, 80S, and polysomes in control

and Kin17 KD 293T cells (n=3). (D) Ratios of 40S to 80S, 60S to 80S, and monosome to polysome were determined in control and Kin17 KD 293T cells (n=3). Statistical analysis was performed using two-tailed Student's *t*-test. ** $p < 0.01$, *** $p < 0.001$; ns no significant. Error bars denote the SEM.

3) The Authors also write "we demonstrate that Kin17 promotes rDNA transcription through recruiting NCL to increase the loading of Pol I complex on the rDNA promoter, consequently promotes ribosomal biogenesis." If the phenotype is to be ascribed to the impact of Kin17 with NCL and/or Pol I, would the loss of NCL and / or Pol I activity perse lead to similar phenotypes as the Kin17 cKO in cortical development?

RESPONSE: Mice homozygous for a knockout of Polr1a, a core subunit of Pol I, exhibit embryonic lethality prior to implantation. Conditional ablation of Polr1a in forebrain progenitors results in a hypoplastic telencephalon and a marked reduction in telencephalon size (PMID:37075751). Similarly, depletion of Polr1a or Polr1c in the neuroepithelium and neural crest cells (NCCs) leads to NCC apoptosis and severe craniofacial abnormalities (PMID:35881792). Notably, NCL depletion produces similar defects in embryonic craniofacial development (PMID:35762670). While there is currently no direct evidence linking Polr1a or NCL to cortical development, these prior findings suggest that their loss could plausibly contribute to cortical atrophy. We have now expanded our discussion to include these previously published findings (page 16 line 495-503).

4) I also recommend the Authors to include the analysis of apoptosis and/or cell death to gain more insights about the fate of the defective neurons in the developing brain.

RESPONSE: As suggested by the Reviewer, we have now assessed apoptosis using TUNEL staining. Our results show that Kin17 cKO mice showed a significantly higher number of TUNEL-positive cells in the cortical brain at E16.5 compared to control mice (Response Document Figure 3A-B), indicating that defective neurons in the developing brain are prone to cell death. We have included these data in the revised manuscript (page 5 line 145-148).

Response Document Figure 3

Response Document Figure 3. (A) Representative images of TUNEL staining (green) and DAPI counterstaining (blue) in the cortex of control and Kin17 cKO mice at E16.5. (B) Quantification of TUNEL-positive cells in the cortices of control and Kin17 cKO ($n = 5$ per group). Statistical analysis was performed using two-tailed Student's t -test; *** $p < 0.001$. Error bars denote the SEM.

5) Finally, it would be important to gain some insights about the impact of loss of Kin17 on the nuclear localization of nucleolin as well as other nucleolar proteins in the Kin17 cKO mouse cortex, to understand whether this results in nucleolar stress, and (also linked to point 4) in increased p53 levels, and apoptosis.

RESPONSE: We have examined the distribution of nucleolar proteins, including nucleolin (NCL) and nucleophosmin 1 (NPM1), by conducting immunostaining in control and Kin17 cKO cortices. We found that both NCL and NPM1 remained localized within the nucleoli in Kin17 cKO cortices (Response Document Figure 4A-B). We have included these data in the revised manuscript (page 5 line 145-148).

Additionally, we have examined p53 protein levels by conducting immunoblotting and detected no difference between control and Kin17 cKO mouse cortices (Response Document Figure 4C). These findings indicate that the apoptosis observed in Kin17

cKO mice occur independently of p53 signaling. We have included these data in the revised manuscript (page 5 line 148-154).

Response Document Figure 4

Response Document Figure 4 (A) Representative immunofluorescence images of NPM1 (green) and DAPI (blue) in cerebral cortex of control and Kin17 cKO mice at E16.5. Scale bar, 50 μm (left) and 5 μm (right). **(B)** Representative immunofluorescence images of NCL (red) and DAPI (blue) in cerebral cortex of control and Kin17 cKO mice at E16.5. Scale bar, 50 μm (left) and 5 μm (right). **(C)** Levels of p53 protein in control and Kin17 KD 293T cells, by immunoblotting and densitometric analysis (n = 3). β-actin was included as a loading control. Statistical analysis was performed using two-tailed Student's *t*-test. ns, no significant change. Error bars denote the SEM.

6) The Authors should also provide some evidence of interaction of Kin17 with NCL not only in the neurospheres, but also in tissue sections.

RESPONSE: As suggested by the Reviewer, we have now examined the interaction of Kin17 and NCL in cortical tissue of WT mice by conducting Co-IP using an anti-Kin17 antibody, followed by immunoblotting detection with an anti-NCL antibody. This assay revealed an association of Kin17 and NCL. To further confirm this interaction, we performed a reciprocal Co-IP using an anti-NCL antibody, followed by immunoblotting detection with an anti-Kin17 antibody (Response Document Figure 5). We have now included these data in the revised manuscript (page 11 line 345-347).

Response Document Figure 5

Response Document Figure 5. The interaction of endogenous Kin17 and NCL in the cortex of WT mice at E16.5, assessed by co-IP using an anti-Kin17 antibody followed by immunoblotting detection with an anti-NCL antibody, and reciprocally by co-IP with an anti-NCL antibody followed by immunoblotting detection with an anti-Kin17 antibody. β -actin was included as a loading control.

We greatly appreciate the Reviewers supportive comments and would again like to take this chance to sincerely offer our gratitude for the helpful guidance about how to improve the scientific rigor of our study.

Reviewer #2:

The study entitled "Kin17 promotes cortical lamination through eliciting rDNA transcription and ribosomal biogenesis in mice" by Li et al. examines the role of Kin17 in cortical lamination. The authors indicate that Kin17 cKO mice display a notable reduction in cortex size and cell numbers. They attempt to link these phenotypic changes to ribosome biogenesis through recruiting NCL and Pol I to the rDNA promoter. It is essential to determine the distinct contributions of NCL and Polr1a in facilitating rDNA transcription and how their interactions with Kin17 enhance ribosomal biogenesis in NPCs. Some experiments were well conducted, but several issues need to be addressed. There are many experiments and data poorly explained, the order of the figures lack logic and most significantly the conclusions are not justified by the data. In the following part my major and minor concerns are listed in more detail.

We thank the Reviewer for the positive comments, as quoted here "Some experiments were well conducted". We provide our full point-by-point responses below.

1. My primary concern is that the depletion of a crucial factor like Kin17 can lead to many functional consequences in general, it is not solely specific to NPCs proliferation and differentiation. It is not clear what cells in the brain express Kin17 and the factor impacts various other cellular processes and functions. What other nerve subtype cells express it? Why then do you think that Kin17 is affecting neurons only? In your papers figures, there are also no clear immunofluorescence staining images demonstrating the expression of Kin17 in neural stem cells.

RESPONSE: By conducting immunofluorescence staining for Kin17 alongside the neuronal marker NeuN or the astrocytic marker GFAP in adult brain tissue, we observed a complete colocalization of Kin17 with NeuN and no colocalization of Kin17 with GFAP (Support Document Figure 6A-B). This indicates that Kin17 is specifically localized to neurons in the mature brain. Moreover, Kin17 is more abundantly expressed in developing brains than compared to adult brains, prompting us to investigate its potential role in neuronal development. We have now included these data in the revised manuscript (page 4 line 94-98).

We have now demonstrated the localization of Kin17 in neural stem cells by conducting immunostaining for Kin17 and neural stem marker nestin in the cortices of E14.5 mice (Support Document Figure 6C). We have now included these data in the revised manuscript (page 4 line 98-101).

Response Document Figure 6

Response Document Figure 6 (A) Representative immunofluorescence images of Kin17 (green), NeuN (red), and DAPI (blue) in 3-month-old WT brain. (B) Representative immunofluorescence images of Kin17 (green), GFAP (red), and DAPI (blue) in 3-month-old WT brain. (C) Representative immunofluorescence images of

Kin17 (green), nestin (red), and DAPI (blue) in the cortex of WT mice at E14.5. Scale bars, 50 μ m.

2. On the one hand depletion of Kin17 in NPCs results in brain atrophy, including NPC proliferation (the number of PH3-positive cells and BrdU-positive cells) and NPC differentiation (*Tbr1*, *Bcl11b*, *Satb2*) at E16.5 stage, but on the other hand, these changes in the proliferation and differentiation of neural stem cells are not observed at the E14 stage. Considering that neural development is a sequential process, and the expression level of Kin17 protein is significantly higher at the embryonic E14 stage compared to the E16 stage (Fig 1A). It is perplexing why the knockout of Kin17 protein affects brain development only at the E16 stage. Furthermore, your experimental data do not support why this phenomenon occurs. What is the molecular mechanism underlying this discrepancy?

RESPONSE: Thank you for focusing our attention on this issue. Although Kin17 expression was most significantly reduced in Kin17 cKO cortices at E14.5 compared to controls, no detectable difference in rRNA expression levels was observed at this stage (Response Document Figure 7A). However, a significant reduction in rRNA expression was detected in Kin17 cKO cortices by E16.5 (Response Document Figure 7B). This may explain the absence of notable defects in cortical lamination and overall brain structure. We have included these data in the revised manuscript (page 8 line 248-251).

Response Document Figure 7

Response Document Figure 7 (A-B) 45S pre-rRNA level was determined in control and Kin17 cKO cortices at E14.5 (A) and E16.5 (B), by qPCR analysis (n=3). Statistical analysis was performed using two-tailed Student's *t*-test. * $p < 0.05$, ns, not significant. ns, no significant change. Error bars denote the SEM.

3. Why did the author choose neurospheres as the material for RNA sequencing? The in vitro culture conditions differ significantly from the in vivo physiological state. Using in vivo tissue materials would better reflect the causes of these differences. It is recommended that the authors perform joint analysis with in vivo tissue sequencing.

RESPONSE: In addition to neurospheres, we also conducted RNA-seq on cortical tissue from control and Kin17 cKO mice at E16.5. By comparing the RNA-seq data from neurosphere and cortical tissue, we identified 1,401 DEGs that were commonly altered in Kin17 cKO versus control samples (Response Document Figure 8A). GO analysis of these shared DEGs revealed enrichment in terms such as “structural constituent of ribosome”, “cytoplasmic translation”, and “cytosolic ribosome” (Response Document Figure 8B). These lines of evidence indicate that Kin17 plays a critical role in regulating ribosome structure and function. We have included these data in the revised manuscript (page 8 line 235-239).

Response Document Figure 8

Response Document Figure 8. (A) Venn diagram showing the number of DEGs in

Kin17 cKO versus control neurosphere, Kin17 cKO versus control E16.5 cortices, and the overlap DEGs between the two datasets. (B) GO analysis of the shared genes, highlighting enriched biological processes.

4. The authors claim that Kin17 binds to rDNA, and then regulate its transcriptional activity, but proper controls for such a statement are missing. Moreover, it is not easy to understand why the authors focused on NCL in their further analyses. The authors should provide more compelling evidence that NCL is actually expressed in NPCs.

RESPONSE: We have now included UBF, a known positive control that binds to the promoter of rDNA (PMID:19717978; PMID:11756560), in ChIP-qPCR analysis (Response Document Figure 9A). To assess the specificity of Kin17's rDNA binding, we have now generated a Kin17 mutant lacking the zinc finger domain—its key DNA-binding motif—and included it as a negative control (Response Document Figure 9B). Our results showed that both Kin17 and UBF bound robustly to the promoter of rDNA, while Kin17 zinc finger mutant (M3) did not show any detectable binding to rDNA (Response Document Figure 9C). We have included these data in the revised manuscript (page 10 line 291-293).

NCL was identified as a Kin17-associated protein by immunoprecipitation using an anti-Kin17 antibody followed by MS analysis. NCL was selected for the further studies because it ranked among the top Kin17 interactors and is known to play a key role in rDNA transcription (PMID: 25225127). We have now included these data in the revised manuscript (page 11 line 340-343).

Co-immunostaining of NCL and Sox2, an NPC marker, revealed colocalization of the two proteins, indicating that NCL is expressed in NPCs (Response Document Figure 9D). We have now included these data in the revised manuscript (page 11 line 343-345).

Response Document Figure 9

Response Document Figure 9. (A) Neurospheres were subjected to ChIP assays using an anti-Kin17 antibody, followed by qPCR-based detection of rDNA (n=3). UBF was included as a positive control. (B) Schematic showing the generation of wild-type Kin17 (Kin17-WT) and the zinc finger domain–deleted mutant (Kin17-M3), each tagged with three Flag epitopes on the N-terminus. (C) N2a cells were transfected with either Kin17-M3 or Kin17-WT plasmids for expression. Relative occupancy of Kin17 at the rDNA promoter was assessed by ChIP-qPCR using an anti-Flag antibody, followed by qPCR detection (n=3). (D) Representative images of NCL (red), Sox2 (green), and DAPI (blue) in cerebral cortex of WT mice at E16.5. Scale bar, Scale bar, 50 μm (left), 12.5 μm (right). Statistical analysis was performed using two-tailed Student's *t*-test. *** $p < 0.001$. ns, no significant change. Error bars denote the SEM.

Minor comments

1. The quality of some figure images (such as Fig 1J) needs to be improved. Furthermore, the patterns of Tbr1 (a specific marker for layer VI) immunofluorescence staining at E16.5 stage differ from that reported in previous studies (for instance Mihalas et al., 2016, Cell Reports 16, 92-105 June 28).

RESPONSE: As suggested by the Reviewer, we have now repeated the immunostaining for Fig 1J and included higher resolutions images in the revised manuscript (Fig 1I, J). The original Tbr1 immunostaining image in Fig 1I exhibited higher background fluorescence, particularly in the marginal zone. To address this, we have now repeated the immunofluorescence staining at E16.5 with additional washing

steps to reduce non-specific fluorescence signals. The newly generated images show a Tbr1 staining pattern in the WT brain that is consistent with previously published (Response Document Figure 10, PMID:12773624).

Response Document Figure 10

Response Document Figure 10 (A) Representative immunofluorescence images of Bcl11b (green) and DAPI (blue) in control and Kin17 cKO mouse cortices. **(B)** Representative immunofluorescence images of Tbr1 (green) and DAPI (blue) in control and Kin17 cKO mouse cortices. Scale bar, 50 μ m.

2. Page 6, "Immunostaining conducted 48 hours post-BrdU injection showed a notable decrease in the number of cells positive for BrdU but negative for Ki67 (BrdU+/Ki67-) in Kin17 cKO mice at E15.5 compared to control mice (Figs 2J and 2K)". The immunofluorescence staining images are not very clear in Fig 2J. There should be double-positive cells (BrdU+/Ki67+) in addition to the negative cells (BrdU+/Ki67-).

RESPONSE: Ki67 serves as a known proliferation marker, with expression initiating at the onset of late G1 and ceasing upon cell cycle exit. BrdU-positive/Ki67-positive (BrdU+/Ki67+) cells reflect those that remain actively in DNA replication, while BrdU-positive/Ki67-negative (BrdU+/Ki67-) cells reflect those that have exited the cell cycle and begun differentiation. The number of BrdU+/Ki67+ and BrdU+/Ki67-cells was significantly reduced in Kin17 cKO cortices (Response Document Figure 11A-C), suggesting impaired proliferation and differentiation. We have now repeated immunofluorescence staining of BrdU and Ki67 and included higher resolution image in the revised manuscript (Figure 2J). We have included these data in the revised manuscript (page 7 line 194-199).

Response Document Figure 11

Response Document Figure 11. (A) Representative immunofluorescence images of BrdU (red), Ki67 (green), and DAPI (blue) in cerebral cortices at 24 hours after BrdU injection. Scale bars, 50 μm. (B-C) Quantitative analysis for BrdU+/Ki67+ and BrdU+/Ki67- cells (n = 5 per group). Statistical analysis was performed using two-tailed Student's *t*-test. Error bars denote the SEM. *** $p < 0.001$. ns, no significant change.

3. It is not clear, why the authors look at NCL, when they correlate Kin17 with rDNA promoter sequence. There is a lack of direct experimental evidence to support the rationale behind this choice.

RESPONSE: NCL was identified as a Kin17-associated protein by immunoprecipitation using an anti-Kin17 antibody followed by mass spectrometry (MS) analysis (Response Document Figure 12). NCL was selected for the further studies because it ranked among the top Kin17 interactors and is known to play a key role in DNA transcription (PMID: 25225127). We have now included these data in the revised manuscript (page 11 line 340-434).

Response Document Figure 12

Response Document Figure 12. Kin17-associated proteins were separated by electrophoresis and visualized by silver staining.

4. The manuscript would benefit from careful reading for grammatical errors. At some stages the manuscript is difficult to read, or does not make sense.

RESPONSE: We have now improved the readability of our manuscript by having it professionally edited for language clarity.

We greatly appreciate the Reviewers supportive comments and would again like to take this chance to sincerely offer our gratitude for the helpful guidance about how to improve the scientific rigor of our study.

Reviewer #3:

The authors study the role of Kin17 in NPC, suggesting that the protein plays an important role in development. They suggest that Kin17 activates rDNA transcription through interaction with Nucleolin, thereby promoting ribosome biogenesis.

In my opinion, the results described require a more detailed functional link, showing the involvement of Kin17 in rDNA regulation. Due to its many functions in replication, repair, splicing, mRNA processing, the effects described in this manuscript could also be indirect. Especially the work with the highly repetitive rDNA genes requires additional controls, as these genes are difficult to analyze in ChIP experiments. Additionally, rRNA transcription is modulated by any kind of changes in the cellular status and by indirect effects that are expected by a factor altering the expression of many genes.

Furthermore, before starting my review of the experimental data, I have to say that the authors willingly left out the known fact that Kin17 plays an important role in ribosome biogenesis. The authors cite the paper 34449532, but only mention the spliceosome interaction. The manuscript 34449532 shows a BioID, revealing the interactors of Kin17, clearly showing the direct interaction with the SSU processome and being involved in 60S maturation. Furthermore, no interaction with Nucleolin was detected in this study. In the light of this knowledge, the novelty has to be re-evaluated, and the controls to distinguish rRNA transcription and processing effects have to be much better controlled than presented here. rRNA processing and transcription of rDNA genes are intimately linked and influence each other.

OVERALL RESPONSE: We have now included both positive (UBF) and negative (Kin17 mutant lacking zinc finger domain) control in our ChIP analyses to confirm the rDNA binding affinity of Kin17. Additionally, we have now ruled out the involvement of Tif1a, a known PolII regulator, in the observed phenotypes. The indicated reference (PMID:34449532) indeed reported an association between Kin17 and the SSU processome, supporting a potential role for Kin17 in rRNA processing. Our study experimentally demonstrates that Kin17 binds to rDNA and promotes its transcription, without affecting rRNA processing. We believe this discrepancy may arise from differences in the experimental systems used: while PMID:34449532 utilized human cell lines such as 293 and HeLa, our study employed cultured mouse neurospheres to investigate the Kin17-associated proteins.

It is also important to note that the BioID approach used in the referenced study, which relies on proximity-dependent biotinylation, may capture not only true interacting proteins but also those in close proximity, potentially leading to false positives—especially when identifying highly abundant proteins (PMID: 31161514). We have now expanded our discussion in the revised manuscript (page 15 line 484-494).

If Kin17 regulates the rRNA genes, the protein should be located in the nucleolus. It would be good to show imaging data with the localization of Kin17 with appropriate nucleolar markers.

RESPONSE: We have now performed immunostaining in neurospheres using anti-Kin17 and anti-NCL antibodies, with NCL serving as a well-established nucleolar marker. We detected a colocalization of Kin17 and NCL, indicating a localization of Kin17 in the nucleolus (Response Document Figure 13). We have included these data in the revised manuscript (page 8 line 240-243).

Response Document Figure 13

Response Document Figure 13. Representative immunofluorescence images of Kin17 (red), NCL (green), and DAPI (blue) in neurospheres. Scale bar, 100 μm (left), 3 μm (right).

If Kin17 functions as an activator of rDNA transcription, it should bind to the active rDNA genes that are not methylated. An experiment, called ChIP-Chop (24162999) can distinguish between binding of the protein to the active (non-methylated) and inactive (methylated) rDNA genes. This would first, show a specific targeting to rDNA and provide a clear link to its suggested activating function.

RESPONSE: As suggested by the Reviewer, we have now performed ChIP-Chop analysis to assess the methylation state of Kin17-bound rDNA chromatin. Using HpaII restriction enzyme digestion, followed by PCR detection with specific primers (Response Document Figure 14A), we observed no significant difference in rDNA methylation between control and Kin17 cKO neurospheres (Response Document Figure 14B), suggesting that Kin17 has no impact on the global methylation state of rDNA.

By conducting chromatin IP using an anti-Kin17 antibody, followed by PCR detection, we found that the proportion of methylated rDNA bound by Kin17 was significantly lower than that present in the input, indicating that Kin17 is preferentially enriched at unmethylated rDNA loci (Response Document Figure 14C). The methylation state of UBF-bound rDNA, which is known to be predominantly unmethylated, was assessed and included as a positive control (Response Document Figure 14C). These newly generated data support the conclusion that Kin17 preferentially associates with unmethylated rDNA to facilitate rDNA transcription. We have now included these data in the revised manuscript (page 10 line 306-318).

Response Document Figure 14

Response Document Figure 14. (A) Schematic illustrating the experimental strategy used to measure CpG methylation at rDNA promoter regions. Mouse rDNA was analyzed by digestion with HpaII restriction enzyme, followed by PCR detection using the indicated primers. Digestion efficiency was measured by PCR amplification of Pbluescript sequences using corresponding primers. (B) Distribution of methylated and unmethylated rDNA in control or Kin17 cKO neurospheres (n=3). (C) Kin17 associated rDNA was obtained by chromatin IP using an anti-Kin17 antibody. The relative abundance of methylated rDNA was determined relative to input. As a positive control, UBF associated rDNA was isolated via chromatin IP using an anti-UBF antibody and the corresponding methylation status was evaluated. The distributions of methylated (grey bars) and unmethylated (orange bars) rDNA copies were plotted accordingly (n=3). Statistical analysis was performed using two-tailed Student's *t*-test. Error bars denote the SEM; *** $p < 0.001$; ns, no significant.

A Chip experiment combined with qPCR, testing only one locus in rDNA, having no positive controls (UBF or TTF) is not sufficient. At least IGS regions of rDNA, enhancer and coding sequences have to be tested with appropriate controls. As mentioned above a ChIP-Chop assay to reveal the co-localization of Kin17 with active genes is also required. By the way, the exact primer sequences, positions within rDNA, DNA fragment length, linearity of qPCR are information and parameters that are not provided in the text and supplemental information. This is required.

RESPONSE: Thank you for drawing our attention to this issue. To investigate the global distribution pattern of Kin17 along the rDNA locus, we have conducted ChIP using an anti-Kin17 antibody, followed by qPCR analyses (ChIP-qPCR) with primers targeting multiple distinct regions of the rDNA, including the enhancer, promoter, 5' external transcribed spacer (5'ETS), 18S, 28S, 5.8S, intergenic spacer (IGS), and transcription termination site (TTS) (Response Document Figure 15A). Our results revealed that Kin17 was predominantly enriched at the rDNA promoter (Response Document Figure 15B). As a positive control, we have included the global distribution pattern of UBF across the rDNA locus, which is known to bind to the enhancer,

promoter, and 5'ETS regions (Response Document Figure 15B). We have now included these data in the revised manuscript (page 10 line 293-297)

In accordance with the Reviewer's suggestion, we have now conducted ChIP-Chop assays to evaluate the methylation status of Kin17-bound rDNA. As demonstrated in Response Document Figure 14 and detailed in our prior response, Kin17 preferentially binds to unmethylated rDNA, which is considered active. These findings further support a role of Kin17 in promoting rDNA transcription. We have now included these data in the revised manuscript (page 10 line 306-318).

We have now included the primer sequences, their positions within the rDNA locus, and the corresponding amplicon sizes in Supplementary Table 1 of the revised manuscript. We have also verified the linearity of the qPCR assay using serially diluted cDNA samples (Response Document Figure 15C).

Response Document Figure 15

Figure for referee with unpublished data and its description has been removed upon request by the authors.

Transfecting the zinc-finger domain mutation of Kin17 that should not bind to DNA anymore would also be a good control to evaluate the binding of Kin17 to rDNA and its transcriptional effects.

RESPONSE: We have generated a Kin17 mutant (Kin17-M3) lacking zinc finger domain and tagged with an Flag epitope at its N-terminus (Response Document Figure 16A). N2a cells were transfected with either Kin17-M3 or WT Kin17, both bearing N-terminal Flag tags, followed by ChIP using an anti-Flag antibody and subsequent qPCR analyses. Kin17-M3 showed a substantial reduction in rDNA occupancy compared to WT Kin17 (Response Document Figure 16B). To evaluate the functional consequence of this mutant, we have measured the levels of 45S pre-mRNA and found that WT Kin17, but not Kin17-M3, promotes the production of 45S, indicating that the zinc-finger domain of Kin17 is essential for its rDNA binding and transcriptional regulatory function (Response Document Figure 16C).

Response Document Figure 16

Response Document Figure 16. (A) Schematic showing the generation of wild-type Kin17 (Kin17-WT) and the zinc finger domain-deleted mutant (Kin17-M3), each tagged with three Flag epitopes on the N-terminus. (B-C) N2a cells were transfected with either Kin17-M3 or Kin17-WT plasmids for expression. (B) Occupancy of Kin17 at the rDNA promoter was assessed by ChIP-qPCR using an anti-Flag antibody, followed by qPCR detection (n=3). (C) 45S pre-rRNA level was determined by qPCR-based analysis. Statistical analysis was performed using two-tailed Student's *t*-test. Error bars denote the SEM. *** $p < 0.001$. ns, no significant.

Furthermore, it was shown that Kin17 interacts with Prmt7 and that protein methylation releases it from chromatin. Including a functional mutant of Kin17 would be important to separate individual biological processes.

RESPONSE: We have now generated a Kin17 mutant (R36K), in which the arginine at position 36 was substituted with lysine, tagging with an HA epitope on the N-

terminus. This mutant is a previously characterized PRMT7-binding-deficient variant (PMID: 24140279; PMID: 34449532). N2a cells were transfected with either HA-tagged WT Kin17 or R36K mutant, along with Flag tagged Prmt7 for expression. The interaction of Kin17 and PRMT7 was assessed by Co-IP using an anti-Flag antibody, followed by immunoblotting with an anti-HA antibody. We demonstrated a robust interaction of PRMT7 and WT Kin17, whereas the R36K mutant showed a significantly reduced association with PRMT7, indicating that methylation at the R36 residue is critical for Kin17-PRMT7 interaction (Response Document Figure 17A).

To assess the functional consequence of this mutant, we have conducted ChIP-qPCR analyses using an anti-HA antibody to evaluate rDNA occupancy. Our results demonstrate that the R36K mutant showed significantly higher rDNA binding compared to WT Kin17 in the presence of exogenous PRMT7 (Response Document Figure 17B). We have further determined the impact of Kin17 R36K mutant on 45S pre-rRNA level and our results revealed that Kin17 R36K mutant showed a significantly higher 45S level compared to WT Kin17 in the presence of exogenous PRMT7 (Response Document Figure 17C), suggesting that unmethylated Kin17 promotes rDNA transcription and production. We have now included these data in the revised manuscript (page 10 line 319-339).

Response Document Figure 17

Response Document Figure 17 (A-C) N2a cells were transfected with either HA-tagged WT Kin17 or R36K mutant, along with Flag tagged Prmt7 for expression. (A) The interaction of Kin17 and PRMT7 was assessed by Co-IP using an anti-Flag antibody, followed by immunoblotting with an anti-HA antibody. (B) rDNA occupancy of Kin17 was determined by ChIP-qPCR analyses using an anti-HA antibody (n=3). (C) 45 pre-rRNA levels, determined by qPCR-based analyses (n=3). Statistical analysis was performed using two-tailed Student's *t*-test. Error bars denote the SEM. *** $p < 0.001$, * $p < 0.05$.

The authors suggest the interaction of Kin17 with Nucleolin through its RNA binding domain. Since the nucleolus is a compartment with high RNA density, and Nucleolin is also an RNA binding protein, it would be important to know whether the interaction is direct or indirectly mediated through RNA. To reveal the specificity of this interaction I recommend to either use purified recombinant proteins to repeat the interaction study, or to include RNase treatment in the IP conditions.

RESPONSE: To determine whether the interaction of Kin17 and NCL is RNA dependent, we have performed Co-IP in the presence or absence of RNase A. We detected a similar extent of Kin17-NCL association in the presence and absence of RNase A, suggesting that the interaction of Kin17 and NCL is independent of RNA (Response Document Figure 18). We have now included these data in the revised manuscript (page 11 line 348-352).

Response Document Figure 18

Response Document Figure 18. Neurosphere lysates were treated with either RNase A or a control solvent, followed by co-IP using an anti-NCL antibody. The NCL-associated proteins were detected by immunoblotting with an anti-Kin17 antibody. Reciprocally, co-IP was performed using an anti-Kin17 antibody, and the Kin17-associated proteins were similarly detected by immunoblotting with an anti-NCL antibody.

The RNA-seq data affects many genes. The authors should survey the effects on known Poll regulators. For example, TIF1A has a similar effect and is embryonic lethal. A downregulation of this factor would have a similar effect.

RESPONSE: By conducting qPCR analysis, we have demonstrated that Tif1a expression levels were unchanged between control and Kin17 cKO neurospheres (Response Document Figure 19A), as well as between control and Kin17 cKO cortical

tissues (Response Document Figure 19B). These findings suggest that Kin17 has no impact on PolII regulator TIF1A.

Response Document Figure 19

Figure for referee with unpublished data and its description has been removed upon request by the authors.

What about the known effects on the replication machinery. The effects would be the same. No proliferation of the cells, as observed. How do the authors rule out this kind of effects. The same with the Ribosome processing effects of Kin17 that were missed to mention. How can the authors be sure that the observed effects are not due to a shutdown of rRNA transcription by defects in rRNA processing?

RESPONSE: By conducting BrdU incorporation in both mice and cultured neurospheres, we demonstrated that Kin17 depletion lead to a reduction in BrdU incorporation, indicating that loss of Kin17 impairs DNA replication (Fig 2H and 2I of the originally submitted manuscript). These results suggest that the observed effect of Kin17 on cell proliferation are at least partially mediated through its role in DNA replication.

We have now examined intermediate rRNA processing products using northern blot analysis. In addition to a reduction in 45S rRNA, we observed a significant decrease in intermediate rRNA products in the cortex of Kin17 cKO mice, including 34S, 32S, and 12S (Response Document Figure 20A). However, the ratio of each intermediate product to the 45S precursor remained comparable between Kin17 cKO and control cortices

(Response Document Figure 20B), suggesting that Kin17 has no impact on rRNA processing. We have included these data in the revised manuscript (page 8 line 256-262).

Response Document Figure 20

Response Document Figure 20 (A-B) Levels of 45S pre-rRNA, intermediate processed product 34S and 12S in control and Kin17 cKO cortices were analyzed by northern blot (A) and densitometric analysis (n=3) (B). U1 was included as a loading control. Statistical analysis was performed using two-tailed Student's *t*-test. Error bars denote the SEM. *** $p < 0.001$. ns, no significant.

We greatly appreciate the Reviewers supportive comments and would again like to take this chance to sincerely offer our gratitude for the helpful guidance about how to improve the scientific rigor of our study.

Dear Dr. Liu,

Thank you for the submission of your revised manuscript to EMBO reports. We have now received the enclosed report from referee 3 who was asked to assess it, and I am happy to say that this referee supports the publication of your study now.

Only a few editorial requests will need to be addressed before we can proceed with the official acceptance of your manuscript:

- Your ms has 5 main figures but the results and discussion sections are not combined. Please either add one more main figure or combine the results and discussion sections to publish your study as a short report. You can find more information about our article types in our guide to authors online.

- Please upload the final ms file as a word file without the figures. All main and EV figures need to be uploaded as individual, high resolution figure files.

- Please add up to 5 keywords to the ms file.

- Please add a Data Availability Section (DAS) after the Methods section, listing a link to data generated in this study and deposited in public databases. If no such data were deposited please mention this fact in the DAS.

- Please add a "Disclosure and Competing Interest Statement" to the ms file.

- Please add all author credits/contributions when you upload the final ms file online.

- The REFERENCE format is not correct, it needs to be alphabetical, not numerical. Please use the EMBO reports reference style.

- Please co-submit a complete author checklist, which you can download from our author guidelines <<https://www.embopress.org/page/journal/14693178/authorguide>>. The completed author checklist will also be part of the transparent peer-review file.

- Please enter all the funders acknowledged in the ms as separate Funders in our online submission system when you upload the final ms.

- Callouts for Fig 2D,J,K,L,M are missing, as well as callouts for Table S3 and S5, please add.

- The 5 suppl. tables should either be part of the Reagents & Tools table or called Expanded View Tables (Table EV1, etc.).

- The 6 suppl. figures should be called Expanded View Figures (Figure EV1, etc). All EV figures need to be uploaded as individual figure files. The legends for the EV figures need to be placed after the main figure legends in the main ms file.

- The Methods section should include a separate Reagents and Tools Table file (listing key reagents, experimental models, software and relevant equipment and including their sources and relevant identifiers). More information on how to adhere to this format as well as downloadable templates (.docx) for the Reagents and Tools Table can be found in our author guidelines: <<https://www.embopress.org/page/journal/14693178/authorguide#manuscriptpreparation>>.

- Materials and Methods should be just Methods.

- During our routine image analysis we detected a potential figure panel reuse between Figure S2I and Figure S6B, which is not listed in the figure legends. Can you please explain/clarify what happened?

Figure Legends - Comments

- Please note that the exact p values are not provided in the legends of figures 1A, C, D, F, L; 2E, G, I, K, L; 3A, C, D, F, H, J; 4C, D, E, F, H, J; 5A, C, F, M-P; S1 G, H; S2 G, S3 C, S5 B, C, F; S6E. Please provide exact values as reasonable.

- Please note that the white borders are not defined in the legend of figure 1I, J, K; 2C, D, F, H, J, M; S2 B, C, D, F, H, I; S3 D, E, G; S6 B. This needs to be rectified.

I would like to suggest to change the ms title. What about:

Kin17 promotes rDNA transcription, ribosomal biogenesis and cortical lamination

EMBO press papers are accompanied online by A) a short (1-2 sentences) summary of the findings and their significance, B) 2-3 bullet points highlighting key results and C) a synopsis image that is exactly 550 pixels wide and 200-600 pixels high (the height is variable). The synopsis image should provide a sketch of the major findings, like a graphical abstract. Please note that text needs to be readable at the final size. Please send us this information along with the final manuscript.

Referee #3:

The authors have thoroughly addressed the comments of the reviewers and included a high number of additional experiments clarifying the raised issues.

The experiments are convincing and the conclusions are sound.

In my opinion the manuscript can be accepted in the present form and is well suited for publication with EMBO Reports.

1) Your ms has 5 main figures but the results and discussion sections are not combined. Please either add one more main figure or combine the results and discussion sections to publish your study as a short report. You can find more information about our article types in our guide to authors online.

Response: We have added one more main figure for the revised manuscript, and results and discussion sections remain independent in the revised manuscript.

2) Please upload the final ms file as a word file without the figures. All main and EV figures need to be uploaded as individual, high resolution figure files.

Response: We have uploaded ms file as a word file without main figures. All main the EV figures have been uploaded individually, with high resolution.

3) Please add up to 5 keywords to the ms file.

Response: We have added 5 keywords in the revised manuscript (page 2 line 44-45).

4) Please add a Data Availability Section (DAS) after the Methods section, listing a link to data generated in this study and deposited in public databases. If no such data were deposited, please mention this fact in the DAS.

Response: We have now added a Data Availability Section to the revised manuscript (page 24 line 772-774). We have deposited the source data for this paper in BioStudies (accession number S-BIAD2101) due to its large file size.

5) Please add a "Disclosure and Competing Interest Statement" to the ms file.

Response: We have now added a "Disclosure and Competing Interest Statement" to the revised manuscript (page 29 line 998-999).

6) Please add all author credits/contributions when you upload the final ms file online.

Response: We have now added author contributions to the revised manuscript (page 29 line 992-997).

7) The REFERENCE format is not correct, it needs to be alphabetical, not numerical. Please use the EMBO reports reference style.

Response: We have now adjusted the REFERENCE format to the EMBO reports reference style.

8) Please co-submit a complete author checklist, which you can download from our

author guidelines <<https://www.embopress.org/page/journal/14693178/authorguide>>.

The completed author checklist will also be part of the transparent peer-review file.

Response: We have now submitted a complete author checklist as instructed.

9) Please enter all the funders acknowledged in the ms as separate Funders in our online submission system when you upload the final ms.

Response: We have now entered all the funders acknowledged in the ms in the online submission system.

10) Callouts for Fig 2D, J, K, L, M are missing, as well as callouts for Table S3 and S5, please add.

Response: We have ensured that all the displaying items are properly cited in the revised manuscript.

11) The 5 suppl. tables should either be part of the Reagents & Tools table or called Expanded View Tables (Table EV1, etc.).

Response: The suppl. Table 3 in the original submission has been incorporated into the Reagent & Tools table, and other suppl. Tables have been renamed to Table EVs.

12) The 6 suppl. figures should be called Expanded View Figures (Figure EV1, etc). All EV figures need to be uploaded as individual figure files. The legends for the EV figures need to be placed after the main figure legends in the main ms file.

Response: We have renamed all suppl. figures to Figure EVs and all EV figures as instructed. We have uploaded these figure files individually and placed the legends for the EV figures after the main figure legends in the revised manuscript.

13) The Methods section should include a separate Reagents and Tools Table file (listing key reagents, experimental models, software and relevant equipment and including their sources and relevant identifiers). More information on how to adhere to this format as well as downloadable templates (.docx) for the Reagents and Tools Table can be found in our author guidelines: <<https://www.embopress.org/page/journal/14693178/authorguide#manuscriptpreparation>>

Response: We have now added a separate Reagents and Tools Table file.

Materials and Methods should be just Methods.

Response: We have renamed Materials and Methods to Methods as instructed.

15) During our routine image analysis, we detected a potential figure panel reuse between Figure S2I and Figure S6B, which is not listed in the figure legends. Can you please explain/clarify what happened?

Response: Thanks for focusing our attention on this issue. The immunostaining experiments for Figure S2I (NCL and DAPI) and Figure S6D (NCL, Sox2, and DAPI) were conducted on the same day. We performed NCL staining first, with or without subsequent Sox2 staining, followed by DAPI counterstaining. The resulting images were captured on the same day, and stored in the same folder. During figure preparation, we inadvertently selected the image intended for Figure S6D (Figure EV6B in revised manuscript) for use in Figure S2I (Figure 2J in the revised manuscript). This error has been corrected in the revised manuscript, and we have carefully reviewed all data to ensure accuracy and proper curation.

Figure Legends - Comments

1) Please note that the exact p values are not provided in the legends of figures 1A, C, D, F, L; 2E, G, I, K, L; 3A, C, D, F, H, J; 4C, D, E, F, H, J; 5A, C, F, M-P; S1 G, H; S2 G, S3 C, S5 B, C, F; S6E. Please provide exact values as reasonable.

Response: We have provided the exact p values in the legends of those above figures.

2) Please note that the white borders are not defined in the legend of figure 1I, J, K; 2C, D, F, H, J, M; S2 B, C, D, F, H, I; S3 D, E, G; S6 B. This needs to be rectified.

Response: The white border indicates the VZ/SVZ in Figure 2F, and the apical surface of VZ in other figures. We have defined the white borders in the legends of these figures (page 30 line 1031-1032; page 31 line 1053-1054; 1055-1056; 1059-1060; 1068-1069; 1073-1074).

3) I would like to suggest to change the ms title. What about:

Kin17 promotes rDNA transcription, ribosomal biogenesis and cortical lamination.

Response: We have changed the title as instructed.

EMBO press papers are accompanied online by A) a short (1-2 sentences) summary of the findings and their significance, B) 2-3 bullet points highlighting key results and C) a synopsis image that is exactly 550 pixels wide and 200-600 pixels high (the height is

variable). The synopsis image should provide a sketch of the major findings, like a graphical abstract. Please note that text needs to be readable at the final size. Please send us this information along with the final manuscript.

Response: We have now included a synopsis text as a separate file, which includes a short summary and highlights.

Additional comments:

Upload Tables EV1-EV3 as separate files (file format: Expanded View Content)

Response: We have now uploaded these tables as instructed.

- Correct 'Supplementary Table EV2' callout in the manuscript text as "Supplementary" is no longer used

Response: We have corrected this in the revised manuscript.

- Please note that ORCID is still missing for Juan Zhang and another request to link ORCID has been sent to the author

Response: We have linked OCRCID for all authors.

- Manuscript sections: remove Author Contributions from the manuscript file (you already provided the credits for each author in the online submission system); place Acknowledgments and Disclosure section after Data Availability section

Response: We have removed Author Contributions from the revised manuscript.

- Data Availability section: provide specific URLs for PRJNA1073761, PRJNA1073873 and S-BIAD2101

Response: We have provided URLs for deposited datasets in the revised manuscript.

- All funders and grants need to be entered in our online system and it appears that just a few have been entered; please add the rest of them separately via More Funders option (please do not use the Comments box): 82071185, 92149303, and 32121002), the Strategic Priority Research Program of the Chinese Academy of Sciences (XDB39000000), CAS Project for Young Scientists in Basic Research (YSBR-013), Plans for Major Provincial Science & Technology Projects (202303a07020004), Research Funds of Center for Advanced Interdisciplinary Science and Biomedicine of IHM (QYZD20220003), the Major Frontier Research Project of the University of Science and Technology of China (LS9100000002), Hefei Comprehensive National Science Center Hefei Brain Project, USTC Research Funds of the Double First-Class Initiative

Response: We have entered all funders in the online system.

- It appears that a callout for Figure 2H is missing in the manuscript text, please add it

Response: We have properly referenced Figure 2H in the revised manuscript.

- Provide the synopsis text as a separate file which has: A) a short (1-2 sentences) summary of the findings and their significance, B) 2-3 bullet points highlighting key results and

- Provide the synopsis image in the following format (not as a PDF): jpeg, TIFF or png format: 550 pixels wide x 200-600 pixels high

Response: We have now included a synopsis text as a separate file, which includes a short summary and highlights. We have now uploaded a synopsis image in a required format.

Dr. Qiang Liu
University of Science and Technology of China
Division of Life Sciences and Medicine
443 Huangshan Road
Anhui 230027
China

Dear Dr. Liu,

I am very pleased to accept your manuscript for publication in the next available issue of EMBO reports. Thank you for your contribution to our journal.

Yours sincerely,
